# Amortized Tree Generation for Bottom-up Synthesis Planning and Synthesizable Molecular Design

**Wenhao Gao[1], Rocío Mercado[1] & Connor W. Coley[1,2]**
[1]Department of Chemical Engineering [2]Department of Electrical Engineering and Computer Science
Massachusetts Institute of Technology
Cambridge, MA 02142, USA
{whgao, rociomer, ccoley}@mit.edu

## Abstract

Molecular design and synthesis planning are two critical steps in the process of molecular discovery that we propose to formulate as a single shared task of conditional synthetic pathway generation. We report an amortized approach to generate synthetic pathways as a Markov decision process conditioned on a target molecular embedding. This approach allows us to conduct synthesis planning in a bottom-up manner and design synthesizable molecules by decoding from optimized conditional codes, demonstrating the potential to solve both problems of design and synthesis simultaneously. The approach leverages neural networks to probabilistically model the synthetic trees, one reaction step at a time, according to reactivity rules encoded in a discrete action space of reaction templates. We train these networks on hundreds of thousands of artificial pathways generated from a pool of purchasable compounds and a list of expert-curated templates. We validate our method with (a) the recovery of molecules using conditional generation, (b) the identification of synthesizable structural analogs, and (c) the optimization of molecular structures given oracle functions relevant to drug discovery.

## 1 Introduction

Designing new functional materials, such as energy storage materials (Hachmann et al., 2011; Janet et al., 2020), therapeutic molecules (Zhavoronkov et al., 2019; Lyu et al., 2019), and environmentally friendly materials (Zimmerman et al., 2020; Yao et al., 2021), is key to many societal and technological challenges and is a central task of chemical science and engineering. However, traditional molecular design processes are not only expensive and time-consuming, but also rely heavily on chance and brute-force trial and error (Sanchez-Lengeling & Aspuru-Guzik, 2018). Thus, a systematic approach to molecular design that can leverage data and minimize the number of costly experiments is of great interest to the field and is a prerequisite for autonomous molecular discovery (Coley et al., 2020a;b).

The core of computer-aided molecular discovery is molecular design. The objective of the task is to identify novel molecules with desirable properties through *de novo* generation or to identify known molecules through virtual screening. There has been a growing interest in applying machine learning methods to tackle this task in recent years (Gómez-Bombarelli et al., 2018; Jin et al., 2018; You et al., 2018; Bradshaw et al., 2019; 2020; Jin et al., 2020; Fu et al., 2021), which has been the subject of many reviews (Elton et al., 2019; Schwalbe-Koda & Gómez-Bombarelli, 2020; Vanhaelen et al., 2020). Despite the large number of models developed, there are few examples that have proceeded to experimental validation or been used in a realistic discovery scenario (Zhavoronkov et al., 2019; Schneider & Clark, 2019). One major barrier to the deployment of these algorithms is that they lack considerations of synthesizability (Gao & Coley, 2020; Huang et al., 2021); Gao & Coley (2020) have demonstrated that when applied to goal-directed optimization tasks, *de novo* molecular design algorithms can propose a high proportion of molecules for which no synthetic plan can be found algorithmically.

Planning and executing a practical synthetic route for a hypothetical molecular structure is a bottleneck that hinders the experimental validation of molecular design algorithms. The goal of computer-assisted synthesis planning (CASP) is to identify a series of chemically plausible reaction steps beginning from available starting materials to synthesize a target chemical compound. Machine learning methods have been applied to improve CASP model performance (Segler et al., 2018; Coley et al., 2018; 2019b; Schwaller et al., 2020; Genheden et al., 2020), and experimental execution has validated significant advances in recent years (Klucznik et al., 2018; Coley et al., 2019c). However, most current algorithms require tens of seconds or minutes to plan a synthetic route for one target compound due to the combinatorial complexity of the tree search. This cost makes a *post hoc* filtering strategy impractical in molecular design workflows that decouple *de novo* design and synthesis planning (Gao & Coley, 2020). However, synthesizability-constrained generation has emerged as a promising alternative to this two-step pipeline (Section 2.1).

In this paper, we report a strategy to generate synthetic pathways as trees conditioned on a target molecular embedding as a means of *simultaneously* addressing the problems of design and synthesis. Proposed pathways are guaranteed to make use of purchasable starting materials and are required to follow the "rules of chemistry" as codified by expert-curated reaction templates, which can be made more or less conservative depending on the application. When applied to synthesis planning, we ask the model to generate synthetic trees conditioned on the target molecule. When applied to synthesizable molecular design, we optimize the fixed-length embedding vector using a numerical optimization algorithm; then, we decode the optimized embedding to obtain the corresponding synthetic tree whose root molecule is the output. The idea builds on the work of Bradshaw et al. (2019) and Bradshaw et al. (2020); however, these methods failed to recover multi-step synthetic paths for any target molecules and were thus only applied to the task of synthesizable analog recommendation. In contrast, the method presented here can successfully recover multi-step retrosynthetic pathways in an amortized manner, in addition to being used for synthesizable analog recommendation.

The main contributions of this paper can be summarized as:

- We formulate a Markov decision process to model the generation of synthetic trees, allowing the generation of multi-step and convergent (i.e., nonlinear) synthetic pathways.
- We propose a model that is capable of (1) rapid bottom-up synthesis planning and (2) constrained molecular optimization that can explore a chemical space defined by a discrete action space of reaction templates and purchasable starting materials.
- We show the first successful attempt to amortized multi-step synthesis planning of complex organic molecules, achieving relatively high reconstruction accuracy on test molecules.
- We demonstrate encouraging results on *de novo* molecular optimization with multiple objective functions relevant to bioactive molecule design and drug discovery.

## 2    RELATED WORK

### 2.1    SYNTHESIZABLE MOLECULAR DESIGN

While most molecular generative models focus on the generation of *valid* molecules with desired properties, there is growing interest in the generation of *synthesizable* molecules, as not all chemically valid molecules are synthetically accessible. MoleculeChef (Bradshaw et al., 2019) was one of the first neural models to cast the problem of molecular generation as the generation of one-step synthetic pathways, thus ensuring synthesizability, by selecting a bag of purchasable reactants and using a data-driven reaction predictor to enumerate possible product molecules. ChemBO (Korovina et al., 2020) extends constrained generation to the multi-step case, but is a stochastic algorithm that generates synthetic pathways iteratively using random selections of reactants as input to another data-driven reaction predictor. While MoleculeChef and ChemBO use neural models for reaction outcome prediction as the ground truth for chemical reactivity (Coley et al., 2019b; Schwaller et al., 2019), reaction templates provide an alternate means of defining allowable chemical steps, algorithmically (Coley et al., 2019a) or by hand-encoding domain expertise (Molga et al., 2019). PGFS (Gottipati et al., 2020) and REACTOR (Horwood & Noutahi, 2020) both use discrete reaction templates and formulate the generation of multi-step synthetic pathways as a Markov decision process and optimize molecules with reinforcement learning. Both are limited to linear synthetic pathways, where intermediates can only react with purchasable compounds and no reaction can occur between

two intermediates. Their inability to design convergent syntheses limits the chemical space accessible to the model. It is worth noting that there also exist previously reported methods for non-neural synthesizability-constrained molecular design, such as SYNOPSIS (Vinkers et al., 2003) and DOGS (Hartenfeller et al., 2012), which pre-date deep molecular generation.

Most recently, Bradshaw et al. (2020) introduced the DoG-AE/DoG-Gen model, which treats synthetic pathways as directed acyclic graphs (DAGs). DoG-Gen serializes the construction of the DAGs and uses a recurrent neural network for autoregressive generation. Dai Nguyen & Tsuda (2021) also employ an autoencoder (AE) framework, jointly trained with a junction tree variational autoencoder (JT-VAE) (Jin et al., 2018). However, none of the previous methods for synthesizable molecular generation have succeeded in achieving high reconstruction accuracy.

## 2.2 SYNTHESIS PLANNING

Algorithms and models for synthesis planning have been in development since the 1960s when retrosynthesis was first formalized (Corey & Wipke, 1969). Various data-driven approaches have been introduced in recent years (Segler et al., 2018; Coley et al., 2018; 2019b; Schwaller et al., 2020; Genheden et al., 2020), although expert methods with human-encoded "rules of chemistry" have arguably achieved greater success in practice (Klucznik et al., 2018; Mikulak-Klucznik et al., 2020). The primary distinction between these methods is how allowable single-step chemical transformations are defined to mimic physical reality as closely as possible; they can all make use of similar tree search algorithms. While these tools can be used to plan routes to target molecules and filter compounds from *de novo* generation for which no pathway is found, none of them can be directly used for molecular generation. Moreover, they all approach synthesis planning *retrosynthetically*, working recursively from the target molecule towards purchasable starting materials (i.e. in a top-down manner), whereas we propose a bottom-up approach that has the potential to be more computationally efficient by mitigating the need for a tree search.

## 2.3 COMBINING SYNTHESIZABLE DESIGN AND SYNTHESIS PLANNING

Our method can be used for synthesizable molecular design and synthesis planning. While our approach is most similar to Bradshaw et al. (2020)'s RetroDoG model, their model was only applied to structural analog generation, and could not demonstrate the successful recovery of target molecules. Bradshaw et al. (2019) showed some examples of successful recovery, but their formulation restricts the search to single-step synthetic pathways, which severely limits its practical utility for both tasks. In contrast, our model can successfully handle multi-step reactions.

## 3 METHOD

### 3.1 PROBLEM DEFINITION

We model synthetic pathways as tree structures called *synthetic trees* (Figure 6A in Appendix A). A valid synthetic tree has one root node (the final product molecule) linked to purchasable building blocks via feasible reactions according to a list of discrete reaction templates. A reaction template is a pattern defining a structural transformation on molecules that is intended to represent a valid chemical reaction, usually encoded as a SMARTS string (Figure 6B&C). We use a list of reaction templates to define feasible chemical reactions instead of a data-driven reaction predictor so that the practical utility of the model can be improved by refining or expanding this set without changing its architecture. Given a list of reaction templates, $\mathcal{R}$, and a list of purchasable compounds, $\mathcal{C}$, our goal is to generate a valid synthetic tree, $T$, that produces a root molecule with a desired structure or function. The product molecule and intermediate molecules in the tree are not themselves generated by the model, but are implicitly defined by the application of reaction templates to reactant molecules. Compared to the generation of molecular graphs, the generation of synthetic trees is more difficult because of the additional constraints of enforcing chemical reaction rules and the commercial availability of starting materials.

**Synthesis Planning** This task is to infer the synthetic pathway to a given target molecule. We formulate this problem as generating a synthetic tree, $T$, such that the product molecule it produces (molecule at the root node), $M_{\text{product}}$, matches the desired target molecule, $M_{\text{target}}$.

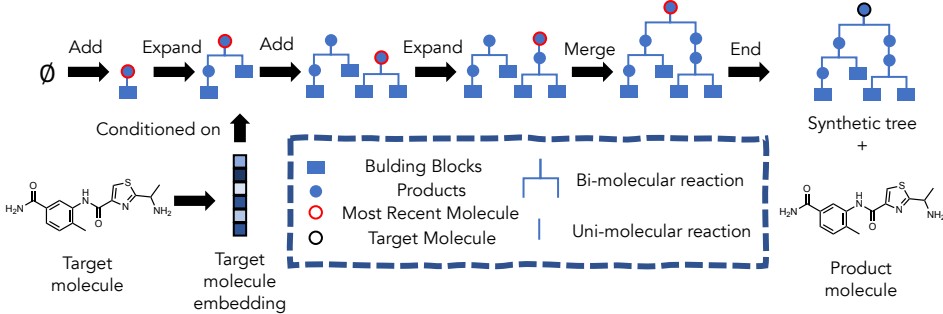

Figure 1: An illustration of the iterative generation procedure. Our model constructs the synthetic tree in a bottom-up manner, starting from the available building blocks and building up to progressively more complex molecules. Generation is conditioned on an embedding for a target molecule. If the target molecule is in the chemical space reachable by our template set and building blocks, the final root molecule should match or at least be similar to the input target molecule.

**Synthesizable Molecular Design** This task is to optimize a molecular structure with respect to an oracle function, while ensuring the synthetic accessibility of the molecules. We formulate this problem as optimizing the structure of a synthetic tree, $T$, with respect to the desired properties of the product molecule it produces, $M_{\text{product}}$.

## 3.2 SYNTHETIC TREE GENERATION AS A MARKOV DECISION PROCESS

We propose an amortized approach to tackle the probabilistic modeling of synthetic trees. In this approach, we model the construction of a synthetic tree as a Markov decision process (MDP), which requires that the state transition satisfies the Markov property: $p(S^{(t+1)}|S^{(t)}, \ldots, S^{(0)}) = p(S^{(t+1)}|S^{(t)})$. This property is naturally satisfied by synthetic trees: upon obtaining a specific compound (an intermediate in a synthetic route), subsequent reaction steps can be inferred entirely from the intermediate compound, and do not depend on the pathway used to get to said compound when conditioned on the target molecule. Below, we first introduce an MDP framework for synthetic tree generation, and then introduce the model to solve it. In our framework, we only allow uni- and bi-molecular reactions.

At a high level, we construct a synthetic tree one reaction step at a time in a bottom-up manner. Figure 1 illustrates a generation process for synthesis planning purposes. We enforce that the generation process happens in a reverse depth-first order, and that no more than two disconnected sub-trees are generated simultaneously.

**State Space** We define the state, $S^{(t)}$, as the root molecule(s) of an intermediate synthetic tree, $T^{(t)}$ at step $t$. Because we enforce that at most two sub-trees can occur simultaneously, there can be at most two root molecules. All root nodes are generated in a reverse depth-first fashion; additionally, we enforce that the synthetic tree always expands from the most recently added node ($M_{\text{most\_recent}}$), and that any merging always happens between two root nodes. The state embedding of a synthetic tree is thus computed by concatenating the embeddings for the two root molecules.

**Action Space** We decompose the action taken at each iteration into four components: (1) the action type, $a_{\text{act}}$, which samples from possible actions "Add", "Expand", "Merge", and "End"; (2) the first reactant, $a_{\text{rt1}}$, which samples from either $\mathcal{C}$ or $M_{\text{most\_recent}}$; (3) the reaction template, $a_{\text{rxn}}$, which samples from $\mathcal{R}$; and (4) the second reactant, $a_{\text{rt2}}$, which samples from $\mathcal{C}$.

(a) If $a_{\text{act}} = $ "Add", one or two new reactant nodes will be added to $T^{(t)}$, as well as a new node corresponding to their product given a specific reaction template. This is always the first action used in building a synthetic tree and leads to an additional sub-tree.

(b) If $a_{\text{act}} = $ "Expand", the most recent molecule, $M_{\text{most\_recent}}$, is used as the first reactant, and a second reactant is selected if $a_{\text{rxn}}$ is a bi-molecular reaction template. If $a_{\text{rxn}}$ is a uni-molecular reaction template, only a new product node is added to $T^{(t)}$. If $a_{\text{rxn}}$ is a bi-molecular reaction template, both a new product node and a new reactant node are added.

(c) If $a_{\text{act}} = $ "Merge", the two root nodes are used as the reactants in a bi-molecular reaction. In this case, a new product node is added to $T^{(t)}$ and the two sub-trees are joined to form one sub-tree.

(d) If $a_{\text{act}} = $ "End", $T = T^{(t)}$ and the synthetic tree is complete. The last product node is $M_{\text{product}}$.

**State Transition Dynamics** Each reaction represents one transition step. To ensure that each reaction step is chemically plausible and has a high likelihood of experimental success, we incorporate domain-specific reaction rules encoded as reaction templates in $\mathcal{R}$. Once a valid action is selected, the transition is deterministic; infeasible actions that do not follow a known template are rejected. Importantly, the structure generated by template application is explicitly incorporated into the new state, whereas the RNN model in Bradshaw et al. (2020) had to implicitly learn the dynamics of the environment (i.e., the outcome of the reaction predictor).

**Reward** For synthesis planning, the reward is the similarity of the product to the target molecule, with a similarity of 1.0 being the highest reward and indicating a perfect match. For molecular design, the reward is determined by how well the product properties match the desired criteria.

## 3.3 Conditional Generation for Synthesis Planning

We model synthesis planning as a conditional synthetic tree generation problem. To solve this MDP, we train a model to predict the action, $a^{(t)}$, based on the state embedding, $z_{\text{state}}^{(t)}$, at step $t$, conditioned on $M_{\text{target}}$. Concretely, at each step, our model, $f$, estimates $a^{(t)} = (a_{\text{act}}^{(t)}, a_{\text{rt1}}^{(t)}, a_{\text{rxn}}^{(t)}, a_{\text{rt2}}^{(t)}) \sim p(a^{(t)}|S^{(t)}, M_{\text{target}})$.

As summarized in Figure 2, our model consists of four modules: (1) an *Action Type* selection function, $f_{\text{act}}$, that classifies action types among the four possible actions ("Add", "Expand", "Merge", and "End"); (2) a *First Reactant* selection function, $f_{\text{rt1}}$, that predicts an embedding for the first reactant. A candidate molecule is identified for the first reactant through a k-nearest neighbors (k-NN) search from the potential building blocks, $\mathcal{C}$ (Cover & Hart, 1967). We use the predicted embedding as a query to pick the nearest neighbor among the building blocks; (3) a *Reaction* selection function, $f_{\text{rxn}}$, whose output is a probability distribution over available reaction templates, from which inapplicable reactions are masked (based on reactant 1) and a suitable template is then sampled using a greedy search; (4) a *Second Reactant* selection function, $f_{\text{rt2}}$, that identifies the second reactant if the sampled template is bi-molecular. The model predicts an embedding for the second reactant, and a candidate is then sampled via a k-NN search from the masked building blocks, $\mathcal{C}'$.

Formally, these four modules predict the probability distributions of actions within one reaction step:

$$
\begin{aligned}
a_{\text{act}}^{(t)} &\sim f_{\text{act}}(S^{(t)}, M_{\text{target}}) = \sigma(\text{MLP}_{\text{act}}(z_{\text{state}}^{(t)} \oplus z_{\text{target}})) \\
a_{\text{rt1}}^{(t)} &\sim f_{\text{rt1}}(S^{(t)}, M_{\text{target}}) = \text{k-NN}_{\mathcal{C}}(\text{MLP}_{\text{rt1}}(z_{\text{state}}^{(t)} \oplus z_{\text{target}})) \\
a_{\text{rxn}}^{(t)} &\sim f_{\text{rxn}}(S^{(t)}, a_{\text{rt1}}^{(t)}, M_{\text{target}}) = \sigma(\text{MLP}_{\text{rxn}}(z_{\text{state}}^{(t)} \oplus z_{\text{target}} \oplus z_{\text{rt1}}^{(t)})) \\
a_{\text{rt2}}^{(t)} &\sim f_{\text{rt2}}(S^{(t)}, a_{\text{rt1}}^{(t)}, a_{\text{rxn}}^{(t)}, M_{\text{target}}) = \text{k-NN}_{\mathcal{C}'}(\text{MLP}_{\text{rt2}}(z_{\text{state}}^{(t)} \oplus z_{\text{target}} \oplus z_{\text{rt1}}^{(t)} \oplus z_{\text{rxn}}^{(t)}))
\end{aligned}
\tag{1}
$$

where $\oplus$ denotes concatenation, $\text{MLP}_*$ denotes a multilayer perceptron (MLP), $z_*$ denotes the embedding of the corresponding entity, $\mathcal{C}'$ denotes a subset of $\mathcal{C}$ that masks out reactants that do not match the selected template $a_{\text{rxn}}^{(t)}$, and k-NN$_{\mathcal{X}}$ is a k-NN from set $\mathcal{X}$. The k-NN search is based on the cosine similarity between the query vector and the embeddings of all molecules in $\mathcal{X}$. Whereas all molecular embeddings ($z_{\text{target}}$, $z_{\text{rt1}}$, and $z_{\text{rt2}}$) are molecular fingerprints (see *Representations* in Section 4.1), $z_{\text{rxn}}^{(t)}$ is a one-hot encoding of $a_{\text{rxn}}^{(t)}$ and $z_{\text{state}}^{(t)}$ is a concatenation of molecular fingerprints for the root molecules in $T^{(t)}$. The $\sigma$ is a softmax masking known invalid actions at each step, such as inapplicable action types (based on the topology of the intermediate tree $T^{(t)}$) and reaction templates (based on the requirement for a subgraph match). Each MLP is trained as a separate supervised learning problem using a subset of information from the known synthetic routes. For further details on the model and algorithm, see Appendices D & E.

## 3.4 Genetic Algorithm for Molecular Optimization

We approach the problem of synthesizable molecular design by optimizing the molecular embedding, $z_{\text{target}}$, on which tree generation is conditioned, with respect to the desired properties in $M_{\text{product}}$. We adopt a genetic algorithm (GA) to perform the numerical optimization on $z_{\text{target}}$. This approach is procedurally simpler than reinforcement learning to solve the MDP and biases generation toward high-performing molecules, enabled by our conditional generation model. The mating pool is defined as a list of molecular embedding vectors and the fitness function is the desired properties of produced

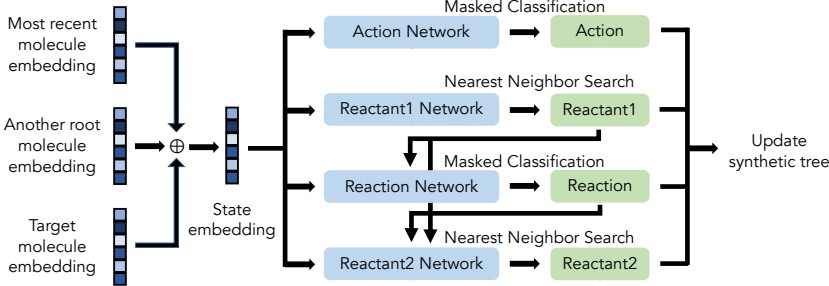

Figure 2: Overview of our model. Within each step, at most two root molecules (most recent and another) and the target molecule as a conditional code fully describe the state. The networks take the embedding of the state and predict the action type, first reactant, reaction template, and second reactant, successively. Those results are used to update the synthetic tree for one reaction step.

product molecules. Within each generation, an offspring pool is constructed from crossover of two vectors randomly sampled from the mating pool as parents. A crossover is defined as inheriting roughly half of the bits from one parent and the remaining bits from another. Mutation can happen to each offspring vector with a small probability, and we decode them to synthetic trees to evaluate the fitness function. The top-performing vectors are selected to form the mating pool for the next generation, and we repeat this process until the stop criteria are met. See Figure 7 for an illustration.

## 4 EXPERIMENTS

### 4.1 EXPERIMENT SETUP

**Reaction Templates** We use a set of reaction templates based on two publicly available template sets from Hartenfeller et al. (2011) and Button et al. (2019). We combine the two sets, removing duplicate and rare reactions, and obtain a final set of 91 reaction templates. This set contains 13 uni-molecular and 78 bi-molecular reactions. 63 are mainly used for skeleton formation, 23 are used for peripheral modifications, and the remaining 5 can be used for either. Within the skeleton formation reactions, we include 45 ring formation reactions, comprising 37 heterocycle formations and 8 carbocycle formations.

**Purchasable Building Blocks** The set of purchasable compounds comprises 147,505 molecules from Enamine Building Blocks (US stock; accessed on May 12, 2021) that match at least one reaction template in our set.

**Dataset Preparation** To prepare a dataset of synthetic pathways obeying these constraints, we applied a random policy to the MDP described in Section 3.2. After filtering by the QED of the product molecules, we obtain 208,644 synthetic trees for training, 69,548 trees each for validation and testing after a random split. We refer to Appendix E for further detail.

**Representations** We use Morgan circular molecular fingerprints of length 4096 and radius 2 to represent molecules as inputs to the MLPs, and Morgan fingerprints of length 256 and radius 2 as inputs to the k-NN module. Additional experiments with other molecular representations showed worse empirical results; further analysis is provided in Appendix I.

**Genetic Algorithm** The GA operates on Morgan fingerprints of 4096 bits and radius 2. The number of bits to inherit is sampled from $\mathcal{N}(2048, 410)$. Mutation is defined as flipping 24 bits in the fingerprint and occurs with probability 0.5. The population size is 128 and the offspring size is 512. The stop criteria is met when either (a) the model reaches 200 generations, or (b) the increase in the population mean value is $< 0.01$ across 10 generations, indicating some degree of convergence.

**Optimization Oracle** To validate our model, we select several common oracle functions relevant to bioactivity and drug discovery, including a docking oracle. The heuristic oracle functions include: quantitative estimate of drug-likeness (QED); octanol-water partition coefficient (LogP); and JNK3, GSK3$\beta$, and DRD2 surrogate models which estimate the response against c-Jun N-terminal kinases-3, glycogen synthase kinase 3$\beta$, and dopamine receptor type 2, respectively. For the docking simulations, we use AutoDock Vina (Trott & Olson, 2010) to dock against the human dopamine receptor $D_3$ (DRD3, PDB ID: 3PBL), and the main protease, $M^{pro}$, of the SARS-Cov-2 virus (PDB ID: 7L11). We access all oracle functions through the Therapeutic Data Common (TDC) interface (Huang et al., 2021).

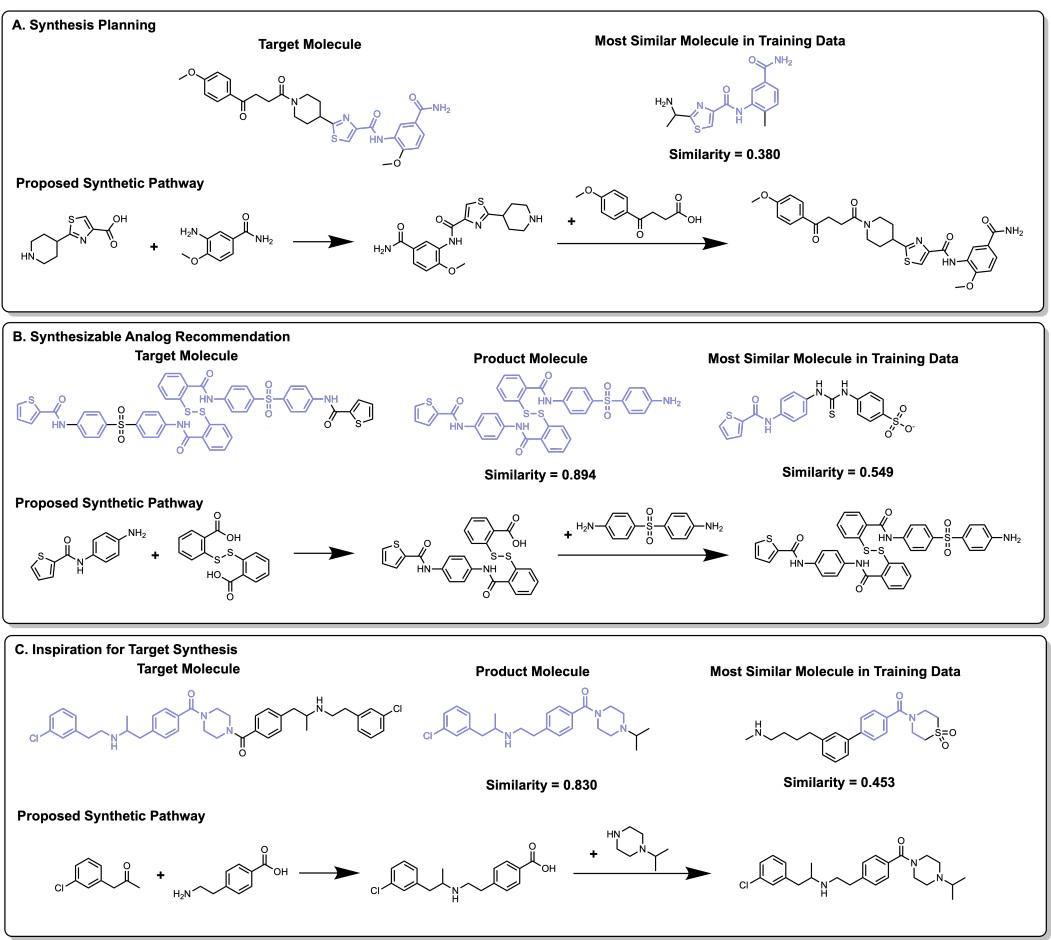

Figure 3: Examples from ChEMBL used as the target molecule for conditioned generation. (A) A successfully recovered molecule where the low similarity to any training examples indicates the generalizability of the model. (B) A molecule that is not recovered as an example of synthesizable analog recommendation (i.e., a similar product). (C) A molecule that is not recovered but may inspire route development to the true target. Matched substructures are highlighted.

## 4.2 SYNTHESIS PLANNING RESULTS

We evaluate the model's ability to reconstruct target molecules that are reachable and unreachable under our specific choice of reaction templates and available building blocks. We use the testing data as "reachable" targets (69,548), and a random sample from ChEMBL as predominantly "unreachable" molecules (20,000). None of the products in the test set are seen in the training and validation stages. We use $k = 3$ in the first reactant k-NN search and expand the trees in a greedy manner ($k = 1$) for each choice. From the obtained product molecules, we choose the one that is the most similar to the target molecule as the output as reflected by the Tanimoto similarity using Morgan fingerprints.

Table 1: Results of synthetic tree construction for "reachable" and "unreachable" target molecules.

| Dataset | N | Recovery Rate↑ | Average Similarity↑ | KL Divergence↑ | FC Distance↓ |
|---|---|---|---|---|---|
| Reachable (test set) | 69,548 | 51.0% | 0.759 | 0.995 | 0.067 |
| Unreachable (ChEMBL) | 20,000 | 4.5% | 0.423 | 0.966 | 1.944 |

Results are summarized in Table 1. The recovery rate measures the fraction of molecules that our model succeeds in reconstructing. Our model can reconstruct 51% of reachable molecules from the held-out test set. As opposed to the typical top-down approach to synthesis planning (i.e., retrosynthesis) that requires tens of seconds or even minutes, our bottom-up approach takes only ~1 second to greedily construct a tree with $k = 1$. Our model only recovers 4.5% of ChEMBL molecules, but we note that a different choice of templates and starting materials can lead to a much higher ChEMBL recovery rate (Gao & Coley, 2020) without changing the model architecture. Figure 3A

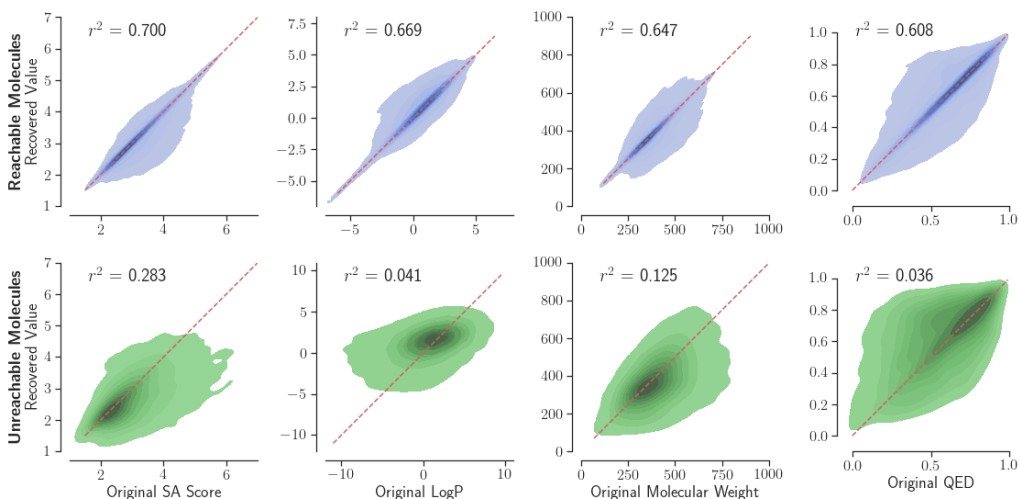

Figure 4: Correlation between properties of $M_{\text{target}}$ and $M_{\text{product}}$ molecules.

shows an example where our model successfully reconstructs a molecule dissimilar to all molecules in the training set. We also assessed the average similarity between target and product molecule pairs, KL divergence (Brown et al., 2019), and Fréchet ChemNet (FC) distance (Preuer et al., 2018) between the targets and recovered products. See Appendix F for additional synthesis planning results.

Among the four networks, the most significant error comes from the first reactant network, $f_{\text{rt1}}$, with a validation accuracy of only 30.8% after k-NN retrieval. To compare, $f_{\text{rt2}}$ reaches a validation accuracy of 70.0% without masking out invalid actions. During sampling, we mask out candidate second reactants that are incompatible with the selected reaction template to achieve a much higher accuracy. The action and reaction networks reach > 99% and 85.8% validation accuracy, respectively.

### 4.3 SYNTHESIZABLE ANALOG RECOMMENDATION RESULTS

We observed that in the cases of unrecoverable molecules, the final products could serve as synthesizable structural analogs under the constraints of $\mathcal{R}$ and $\mathcal{C}$ (see Figure 3B for an example). The metrics in Table 1 show how the product molecules in the unrecovered cases are still structurally similar to the input molecules. We illustrate the correlation between the properties of target and product molecules in Figure 4. We investigated the SA Score (Ertl & Schuffenhauer, 2009b), QED, CLogP, and molecular weight, and observed that most product properties have a positive correlation with the corresponding input properties. The least successful case is QED due to its high sensitivity to structural changes as quantified by the structure-activity landscape index (SALI) (Guha, 2012) (Table 6 in Appendix G). Overall, our model can suggest reasonable synthesizable analogs for target molecules, especially when the desired property is highly correlated with molecular structure.

In other cases where the target product is not recovered by the generated pathway, the output synthetic tree may still provide inspiration for the synthesis of target molecules. Figure 3C highlights an example where the failure of reconstruction is due to the binary Morgan fingerprint's inability to distinguish repeating units. Our model successfully constructed one side of this symmetric molecule. In this case, a synthetic chemist would likely recognize that replacing 1-isopropylpiperazine (the reactant added in the second step) with piperazine may lead to synthesis of the target molecule.

### 4.4 SYNTHESIZABLE MOLECULAR OPTIMIZATION RESULTS

To assess the optimization ability of our algorithm, we first consider common heuristic oracle functions relevant to bioactivity and drug discovery (Tables 2 and 7). Note that the baseline methods we compare to do not constrain synthesizability, which means they explore a larger chemical space and are able to obtain molecules that score higher but are not synthesizable. The results show that our model consistently outperforms GCPN (You et al., 2018) and MolDQN (Zhou et al., 2019), and is comparable to GA+D (Nigam et al., 2019) and MARS (Xie et al., 2021) across different tasks. We highlight the case of GSK3$\beta$ inhibitor optimization in Figure 5. In this task, our model proposes a molecule scored marginally worse than DST (Fu et al., 2021) and MARS, but much

Figure 5: Results of synthesizable molecular design. (A) A comparison between results of our model, DST, MARS, and GA+D on GSK3$\beta$ bioactivity optimization. Our model proposes a highly scored molecule with a much simpler structure than the other baselines. (B) The results of docking score optimization for M$^{\text{pro}}$ of SARS-Cov-2. Our model successfully proposes multiple molecules with stronger predicted binding affinity than a known inhibitor.

simpler in structure. Indeed, this molecule can be accessed within one reaction step from purchasable compounds in $\mathcal{C}$ through a simple Suzuki reaction (see Figure 12 for the synthetic pathway). This makes our model's recommendation more immediately actionable, i.e., ready for experimental validation. As an additional quasi-realistic application to structure-based drug design, we optimized the binding affinity to DRD3 and M$^{\text{pro}}$ of SARS-Cov-2 as an example target protein and successfully generated multiple molecules with improved docking scores relative to a known inhibitor (see Figure 5). We used the Guacamol filter (Brown et al., 2019) and SA_Score (Ertl & Schuffenhauer, 2009a) to quantitatively evaluate the top-100 generated molecules against DRD3 and compared them to the TDC generative benchmark (Huang et al., 2021). Our method is the only one which achieved a high passing rate and low SA_Score, indicating a better quality of the structures (see Table 8). Additional molecular optimization results are available in Appendix H.

Table 2: Highest scores of generated molecules for various *de novo* molecular design tasks. Note that all baselines other than our method don't place constraints on synthetic accessibility.

| Method | JNK3 | | | GSK3$\beta$ | | | QED | | |
|---|---|---|---|---|---|---|---|---|---|
| | 1st | 2nd | 3rd | 1st | 2nd | 3rd | 1st | 2nd | 3rd |
| GCPN | 0.57 | 0.56 | 0.54 | 0.57 | 0.56 | 0.56 | **0.948** | 0.947 | 0.946 |
| MolDQN | 0.64 | 0.63 | 0.63 | 0.54 | 0.53 | 0.53 | **0.948** | **0.948** | **0.948** |
| GA+D | 0.81 | 0.80 | 0.80 | 0.79 | 0.79 | 0.78 | - | - | - |
| MARS | 0.92 | 0.91 | 0.90 | **0.95** | 0.93 | 0.92 | **0.948** | **0.948** | **0.948** |
| DST | **0.97** | **0.97** | **0.97** | **0.95** | **0.95** | **0.95** | 0.947 | 0.946 | 0.946 |
| Our Method | 0.80 | 0.78 | 0.77 | 0.94 | 0.93 | 0.92 | **0.948** | **0.948** | **0.948** |

## 5 CONCLUSION

In this work, we have introduced an amortized approach to conditional synthetic tree generation which can be used for synthesis planning, synthesizable analog recommendation, and molecular design and optimization. Our model bridges molecular generation and synthesis planning by coupling them together into one rapid step, eliminating the need for two-stage pipelines of generation and filtering. We have demonstrated promising results for a variety of molecular optimization tasks with relevance to drug discovery, illustrating how this approach can be used to effectively explore synthesizable chemical space in pursuit of new functional molecules. Additional outlook is in Appendix C.

## REPRODUCIBILITY STATEMENT

The code repository is given in the supplementary material, including instructions in a README file, all code used for data preprocessing, and all code used to train and evaluate the model. All the data we use are either publicly available or can be calculated by open-sourced software. Section 4.1 and Appendix E describe the experimental setup, implementation details, datasets used, and hardware configuration.

## ACKNOWLEDGMENTS

This research was supported by the Office of Naval Research under grant number N00014-21-1-2195. RM received additional funding support from the Machine Learning for Pharmaceutical Discovery and Synthesis consortium. We thank John Bradshaw for helpful discussions. We also thank Tianfan Fu, Samuel Goldman and Itai Levin for commenting on the manuscript.

## CODE AND DATA AVAILABILITY

All code and releasable data can be found at `https://github.com/wenhao-gao/SynNet`. Additional results can be found in the supporting information.

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

APPENDIX

# A  SYNTHETIC TREE AND REACTION TEMPLATE

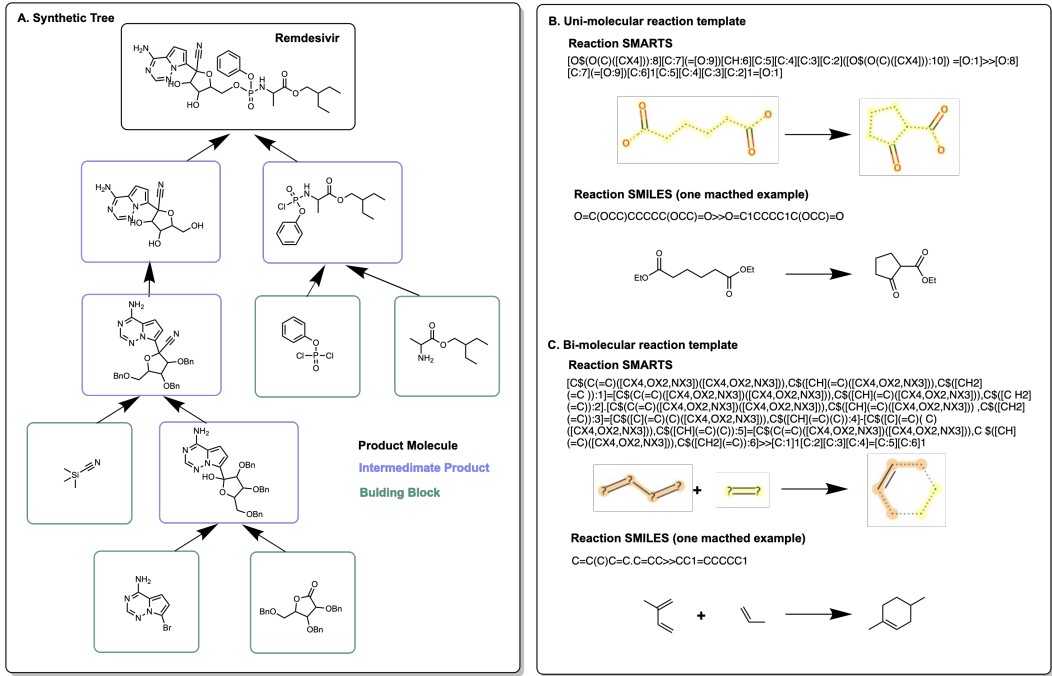

Figure 6: Illustration of a synthetic pathway as a synthetic tree (A) and reaction templates (B & C). (A) is the synthetic tree of remdesivir, a drug authorized for emergency use to treat COVID-19. Different color box labels indicate different types of chemical nodes. (B) and (C) are examples of reaction templates for uni- and bi-molecular reactions, where SMARTS is a specific syntax for encoding reaction transforms.

# B ILLUSTRATION OF GENETIC ALGORITHM

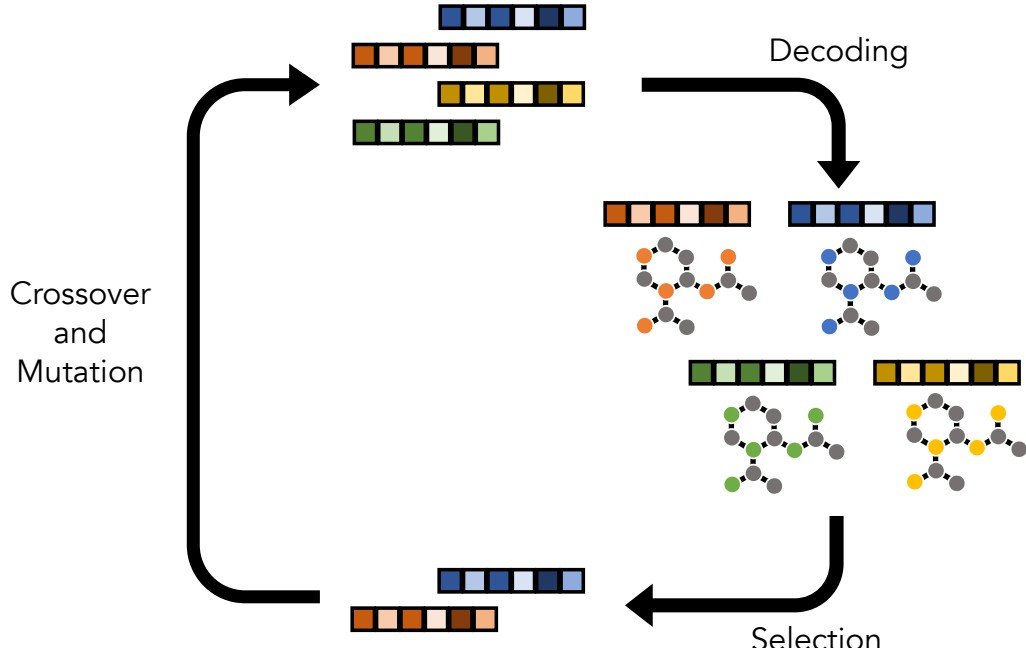

Figure 7: Illustration of the genetic algorithm used to optimize the molecules. We use the conditional synthetic tree generator as a decoder to obtain molecules corresponding to input fingerprints. We crossover and mutate on the pool of fingerprints to optimize the molecule implicitly.

## C  OUTLOOK

While our model shows promising results on the recovery of molecules from generated trees and *de novo* molecular optimization performance, there remain multiple challenges to be addressed:

**Initial Reactant Selection** Our results show that the first reactant selection is the primary bottleneck to target molecule recovery. We consider two major reasons for this: (1) the input to $f_{rt1}$ is usually the target molecule only, with no action mask applied. The large action space (i.e., all purchasable molecules) and the limited input information thus make the problem the most difficult of the four tasks; (2) during the generation of the synthetic trees, we adopt a depth-first approach that implicitly introduces a canonical ordering during reactant selection. The implicit ordering is arbitrary and might harm the training of the network. Allowing the model to predict any valid order or including multiple orders as a form of data augmentation may improve performance. Further, it may be possible to first select the action (template) to constrain the space of compatible first reactants, which may improve performance given that this is a smaller search space.

**Reaction Templates and Purchasable Compound Selection** The definition of *synthesizable* used in this work is that the molecule can be reached with a synthetic pathway only containing starting materials belonging to our list of purchasable compounds and reactions that follow our list of reaction templates. Here, we use only 91 templates, which pales in comparison to the $\sim$1,300 reaction families defined by NameRxn (Sayle & Lowe) the tens of thousands of templates in the expert CASP program SYNTHIA (Szymkuć et al., 2016), and the hundreds of thousands in the data-driven CASP program ASKCOS (Coley et al., 2019c). Using a more comprehensive set of reaction templates would enlarge the chemical space our model can explore. Additional constraints on template applicability could improve the feasibility of pathways, e.g., to mitigate selectivity concerns (cf. Figure 12).

**Molecular Representation and Molecular Similarity** Certain results (e.g., Figure 3C) reveal a limitation of using boolean Morgan fingerprints. As this type of fingerprint only accounts for the presence or absence of specific substructures, it cannot distinguish molecules with different numbers of repeated units or other symmetries. Applying a count or summation-based representation, coupled with development of a similarity measurement based on that representation, would solve this problem.

**Beam Search for Decoding** Besides tackling the aforementioned challenges, in future work we also plan to implement beam search with our networks to enhance the model performance for bottom-up synthesis planning. Improving the sample efficiency of the synthesizable molecular design algorithm (the GA module), and applying it to more high-fidelity computational oracles would also be of interest. Those advances could enable faster and more accurate synthesis planning, as well as an even better tool for *de novo* molecular design.

# D ALGORITHM

---

**Algorithm 1** `Synthetic tree generation`

---

1: **Input**: List of reaction templates $\mathcal{R}$, List of building blocks $\mathcal{C}$, an input molecule $M_{\text{target}}$, molecular encoder $z_M = E(M)$, an action network $\text{MLP}_{\text{act}}$, a reactant1 selection network $\text{MLP}_{\text{rt1}}$, a reaction network $\text{MLP}_{\text{rxn}}$, and a reactant2 selection network $\text{MLP}_{\text{rt2}}$.

2: **Output**: A rooted binary tree $T$, representing a synthesis pathway.

3: Encode target molecule, $z_{\text{target}} \leftarrow E(M_{\text{target}})$

4: Initialize state $S \leftarrow \varnothing$, tree $T \leftarrow \varnothing$, $M_{\text{most\_recent}} \leftarrow None$

5: **for** $t = 1, 2, \cdots, t_{max}$ **do**

6:     Predict and sample action type:
        $a_{\text{act}} \sim p_{\text{act}} = \sigma(\text{MLP}_{\text{act}}(z_{\text{state}} \oplus z_{\text{target}}))$

7:     Predict first reactant, $a_{\text{rt1}} = \text{MLP}_{\text{rt1}}(z_{\text{state}} \oplus z_{\text{target}})$

8:     **if** $a_{\text{act}} = end$ **then**

9:         break

10:     **else if** $a_{\text{act}} = add$ **then**

11:         $M_{\text{rt1}} \leftarrow k - NN_{\mathcal{C}}(a_{\text{rt1}})$

12:         $z_{\text{rt1}} \leftarrow E(M_{\text{rt1}})$

13:     **else**

14:         $M_{\text{rt1}} \leftarrow M_{\text{most\_recent}}$

15:         $z_{\text{rt1}} \leftarrow E(M_{\text{most\_recent}})$

16:     **end if**

17:     Predict and sample reaction template:
        $a_{\text{rxn}} \sim p_{\text{rxn}} \leftarrow \sigma(\text{MLP}_{\text{rxn}}(z_{\text{state}} \oplus z_{\text{target}} \oplus z_{\text{rt1}}))$

18:     **if** $a_{\text{rxn}}$ is $bi\text{-}molecular$ **then**

19:         **if** $a_{\text{act}} = merge$ **then**

20:             $M_{\text{rt2}} \leftarrow S/M_{\text{rt1}}$

21:             $z_{\text{rt2}} \leftarrow E(M_{\text{rt2}})$

22:         **else**

23:             Predict second reactant, $a_{\text{rt2}} \leftarrow \text{MLP}_{\text{rt2}}(z_{\text{state}} \oplus z_{\text{target}} \oplus z_{\text{rt1}} \oplus a_{\text{rxn}})$

24:             $M_{\text{rt2}} \leftarrow k - NN_{\mathcal{C}'}(a_{\text{rt2}})$

25:             $z_{\text{rt2}} \leftarrow E(M_{\text{rt2}})$

26:         **end if**

27:     **end if**

28:     rxn_tem $\leftarrow \mathcal{R}[a_{\text{rxn}}]$

29:     Run reaction $M_{\text{product}} \leftarrow$ rxn_tem$(M_{\text{rt1}}(, M_{\text{rt2}}))$

30:     Update $T$, $S$ and $M_{\text{most\_recent}} \leftarrow M_{\text{product}}$

31: **end for**

---

## E    ADDITIONAL EXPERIMENTAL DETAILS

**Network Setup** For all experiments reported in this paper, the four networks, $f_{act}$, $f_{rt1}$, $f_{rxn}$, and $f_{rt2}$, use 5 fully connected layers with 1000, 1200, 3000, and 3000 neurons in the hidden layers, respectively. Batch normalization is applied to all hidden layers before ReLU activation. In $f_{act}$ and $f_{rxn}$, a softmax is applied after the last layer and cross entropy loss is used, while $f_{rt1}$ and $f_{rt2}$ use a linear activation in the last layer and mean squared error (MSE) loss. We use the Adam optimizer to train all networks with a learning rate of 1e-4 and mini-batch size of 64.

**Training** Each MLP is trained as a separate supervised learning problem using a subset of information from the known synthetic routes. For instance, $f_{rxn}$ is a classification network which learns to select a discrete action given information on the current state of the tree, the target molecule, and the first reactant. Similarly, $f_{act}$ is a classification network which learns to select the correct action type given the state of the tree. On the other hand, $f_{rt1}$ and $f_{rt2}$ learn embeddings (regression) for the first and second reactant candidates, respectively, followed by a nearest-neighbors search from $\mathcal{C}$ and $\mathcal{C}'$.

**Dataset Preparation** Following the procedure we described in Section 3.2, we applied a random policy to generate the synthetic trees. We randomly sampled purchasable reactants, and randomly applied matching reaction templates to them. Doing so, we obtained 550k synthetic trees and filtered by the QED of the product molecules (QED > 0.5) as well as randomly with a probability 1 - QED/0.5 to increase their drug likeness, in a crude sense. Ultimately, we obtained 208,644 synthetic trees for training, 69,548 trees for validation, and 69,548 trees for testing after a random split.

**Docking Procedure** We downloaded the crystal structure of the $M^{pro}$ of SARS-Cov-2 virus with PDB ID: 7L11. We removed the water molecules and ions from the file, and estimated the docking box based on the docked pose with a reported inhibitor, compound 5 in Zhang et al. (2021). For each ligand generated by our method, we use RDKit to generate molecular conformations (Landrum; Riniker & Landrum, 2015) and perform docking simulations using AutoDock Vina (Trott & Olson, 2010). We set exhaustiveness as 8 during the generation and recorded the high-scored conformations for a rescoring with exhaustiveness as 32. The reported values and rank are based on the rescoring.

**Hardware** Models were trained on a node with double Intel Xeon Gold 6230 20-core 2.1GHz processors, 512 GB DDR4 RAM, and eight RTX 2080Ti graphics cards with 11GB VRAM. Data preprocessing and predictions were made on a CPU node with 512 GB RAM and two AMD EPYC 7702 64-core 2GHz processors.

# F    ADDITIONAL RESULTS ON SYNTHESIS PLANNING

Table 3: Results of synthesis planning for molecules from training, validation, and test datasets. Recovery rate indicates the fraction of final product molecules which match the input target molecules. Average similarity measures the Tanimoto similarity between the fingerprints of the product molecules and target molecules.

| Dataset | Recovery Rate↑ | Average Similarity↑ | Average Similarity (Unrecovered)↑ |
|---------|---------------|---------------------|-----------------------------------|
| Train. | 92.1% | 0.975 | 0.688 |
| Valid | 51.4% | 0.761 | 0.508 |
| Test | 51.0% | 0.759 | 0.508 |
| ChEMBL | 4.50% | 0.423 | 0.396 |

Table 4: Analysis of the generated molecules in unrecovered cases. We compare with baseline generative methods to evaluate the similarity of input and output molecules. Baseline data from Guacamol (Brown et al., 2019)

| | Validity↑ | Uniqueness↑ | Novelty↑ | KL Divergence↑ | FCD↑ |
|---|-----------|-------------|----------|----------------|------|
| Random Sample | 1.000 | 0.997 | 0.000 | 0.998 | 0.929 |
| SMILES LSTM | 0.959 | 1.000 | 0.912 | 0.991 | 0.913 |
| AAE | 0.882 | 1.000 | 0.998 | 0.886 | 0.526 |
| VAE | 0.870 | 0.999 | 0.974 | 0.982 | 0.863 |
| Our Model (Reachable) | 1.000 | 0.999 | 1.000 | 1.000 | 0.920 |
| Our Model (Unreachable) | 1.000 | 0.988 | 1.000 | 1.000 | 0.684 |

# G  ADDITIONAL RESULTS ON ANALOGS RECOMMENDATION

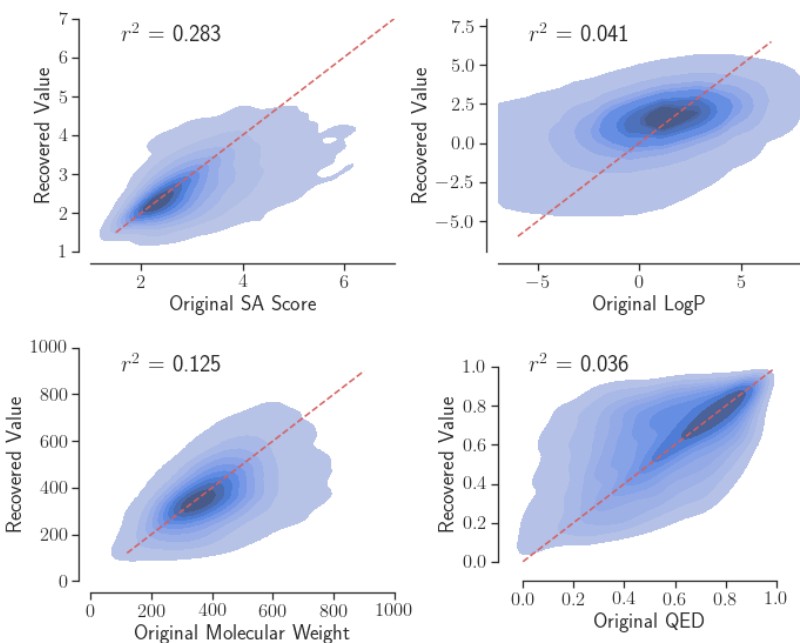

Figure 8: Correlation between input and output values on ChEMBL molecules.

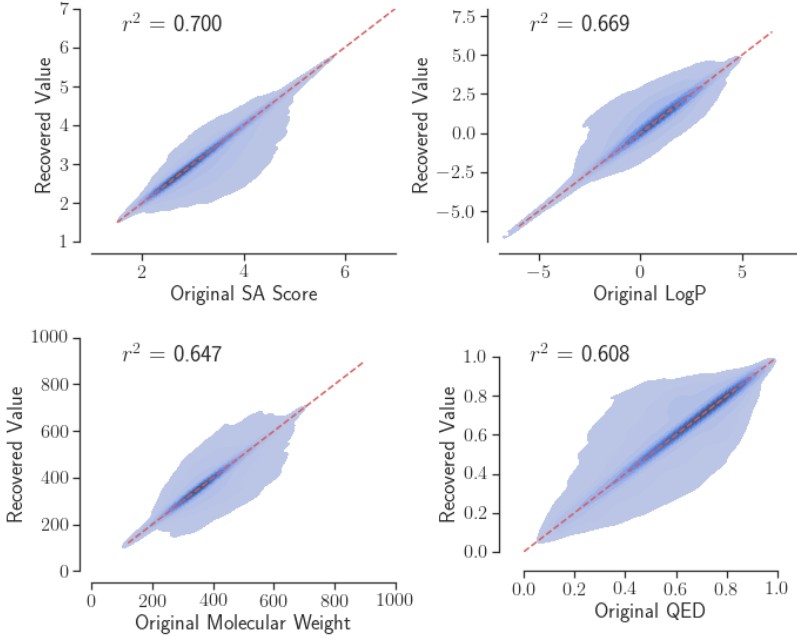

Figure 9: Correlation between input and output values on test set molecules.

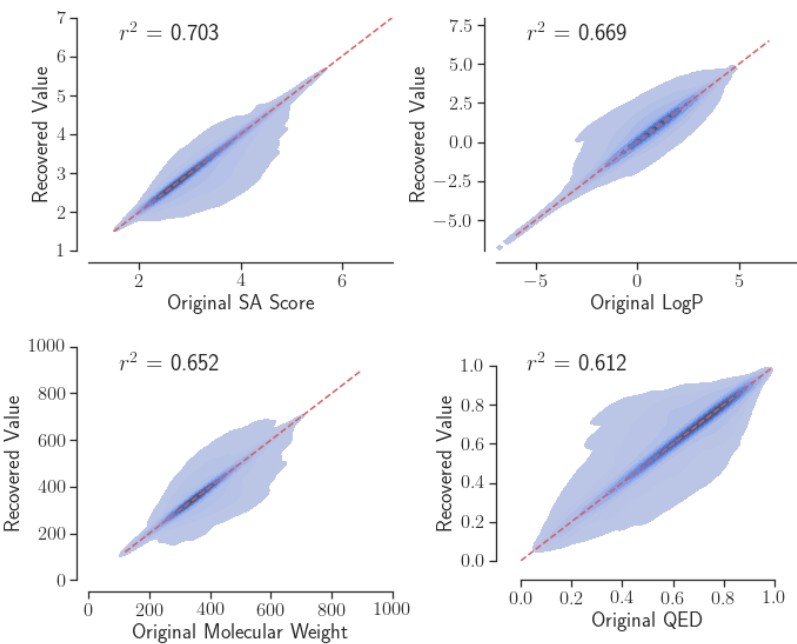

Figure 10: Correlation between input and output values on validation set molecules.

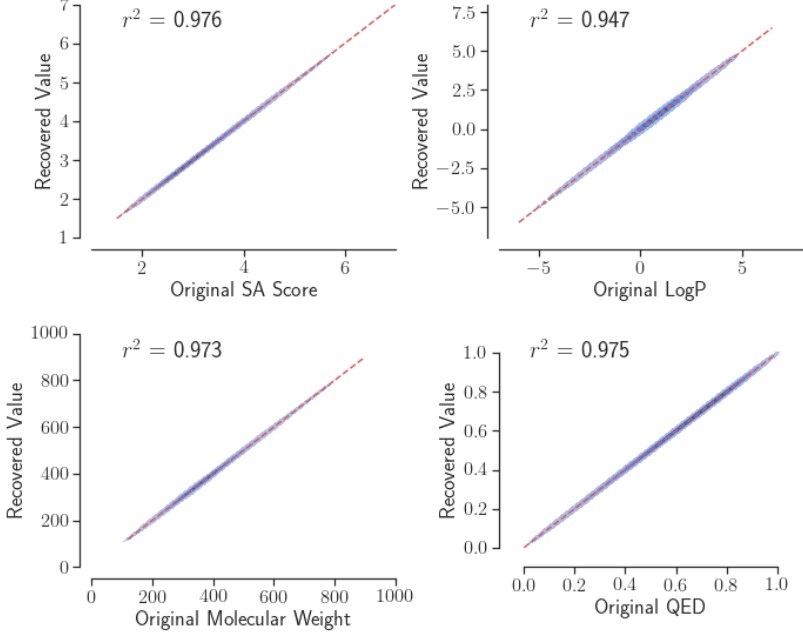

Figure 11: Correlation between input and output values on training set molecules.

$$SALI = \frac{1}{N} \sum_{(i,j)} \frac{|d_i - d_j|/range}{1 - sim(i,j)} \qquad (2)$$

Table 5: Results of synthesis planning for unrecovered cases in "reachable" and "unreachable" molecules. Metrics other than recovery rate are measured for unrecovered molecules only.

| Dataset | N | Average Similarity↑ | KL Divergence↑ | FC Distance↓ |
|---|---|---|---|---|
| Reachable (test set) | 69,548 | 0.508 | 1.000 | 0.315 |
| Unreachable (ChEMBL) | 20,000 | 0.396 | 1.000 | 2.140 |

Table 6: SALI of oracle functions we investigated. A higher SALI value means that a small structure change can lead to a larger property change, such that the function is more sensitive to detailed structural changes.

| | SA Score | QED | LogP | Mol. Weight |
|---|---|---|---|---|
| Reachable | 0.139 | 0.368 | 0.053 | 0.127 |
| Unreachable | 0.115 | 0.383 | 0.013 | 0.043 |

## H    ADDITIONAL RESULTS ON MOLECULAR DESIGN

Table 7: Results of synthesizable molecular optimization on common oracle functions. Seeds are randomly sampled from the ZINC database (Sterling & Irwin, 2015). Top-n is the average value for the top n molecules. "Seeds" refers to the mean scores of the initial mating pool we sampled and "Outputs" refers to the mean scores of the 128 generated molecules.

| | Best from seeds | Top-1 | Top-10 | Top-100 | Seeds | Outputs |
|---|---|---|---|---|---|---|
| QED | 0.947 | 0.948 | 0.948 | 0.947 | 0.673±0.289 | 0.946±0.001 |
| LogP | 3.81 | 25.82 | 25.05 | 23.96 | 1.09±35.18 | 23.72±0.69 |
| JNK3 | 0.120 | 0.800 | 0.758 | 0.719 | 0.032±0.025 | 0.715±0.017 |
| GSK3$\beta$ | 0.310 | 0.940 | 0.907 | 0.815 | 0.050±0.051 | 0.803±0.041 |
| DRD2 | 0.181 | 1.000 | 1.000 | 0.998 | 0.007±0.018 | 0.996±0.003 |

Table 8: **Leaderboard on TDC DRD3 docking benchmark using ZINC and Docking.** Mean and standard deviation across three runs are reported. Arrows (↑, ↓) indicate the direction of better performance. The best method is bolded and the second best is underlined. Note in particular the low SA_Score and high % Pass, which are heuristics for synthetic complexity and drug likeness/quality.

| Method Category | | | Domain-Specific Methods | | State-of-the-Art Methods in ML | | | | Ours |
|---|---|---|---|---|---|---|---|---|---|
| Metric | Best-in-data | # Calls | Screening | Graph-GA | LSTM | GCPN | MolDQN | MARS | SynNet |
| Top100 (↓) | -12.080 | | -10.542±0.035 | **-14.811±0.413** | -13.017±0.385 | -10.045±0.226 | -8.236±0.089 | -9.509±0.035 | -11.133 |
| Top10 (↓) | -12.590 | | -11.483±0.056 | **-15.930±0.336** | -14.030±0.421 | -11.483±0.581 | -9.348±0.188 | -10.693±0.172 | -12.020 |
| Top1 (↓) | -12.800 | | -12.100±0.356 | **-16.533±0.309** | -14.533±0.525 | -12.300±0.993 | -9.990±0.194 | -11.433±0.450 | -12.300 |
| Diversity (↑) | 0.864 | 5000 | 0.872±0.003 | 0.626±0.092 | 0.740±0.056 | **0.922±0.002** | 0.893±0.005 | 0.873±0.002 | 0.821 |
| Novelty (↑) | - | | - | 1.000±0.000 | 1.000±0.000 | 1.000±0.000 | 1.000±0.000 | 1.000±0.000 | 1.000 |
| %Pass (↑) | 0.780 | | 0.683±0.073 | 0.393±0.308 | 0.257±0.103 | 0.167±0.045 | 0.023±0.012 | 0.527±0.087 | **0.800** |
| Top1 Pass (↓) | -11.700 | | -10.100±0.000 | **-14.267±0.450** | -12.533±0.403 | -9.367±0.170 | -7.980±0.112 | -9.000±0.082 | -12.300 |
| SA_Score (↓) | 2.973 | | 3.036±0.014 | 4.783±1.195 | **2.611±0.238** | 6.843±0.210 | 6.687±0.049 | 3.103±0.011 | 2.801 |

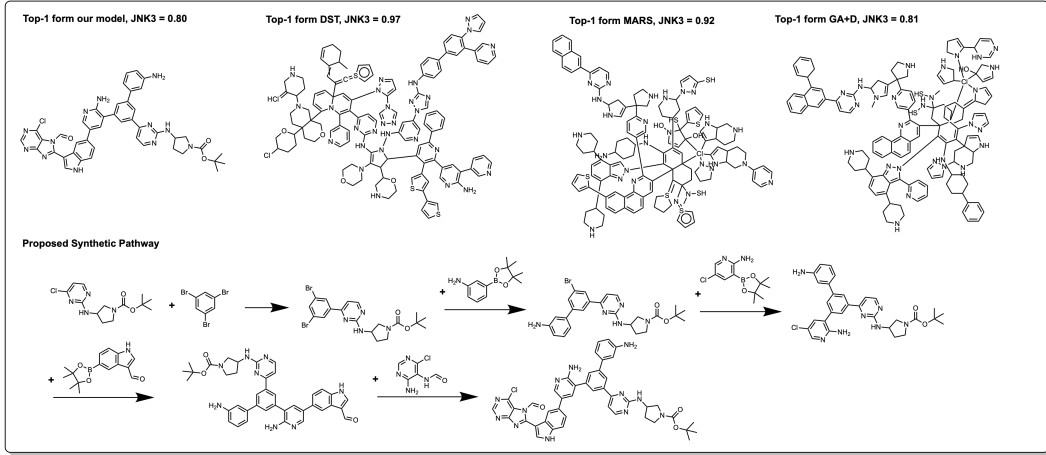

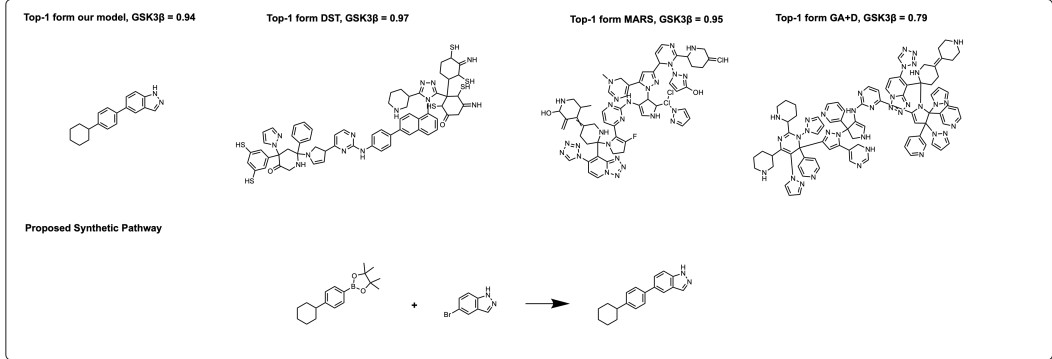

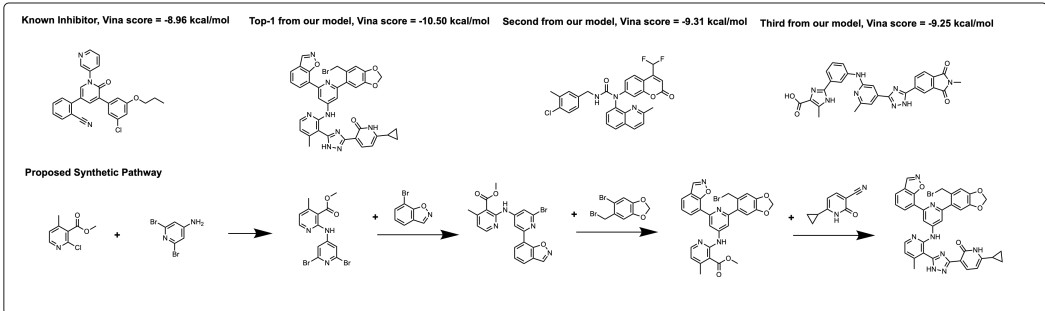

Figure 12: The optimized structures and their corresponding synthetic pathways for various inhibitor design tasks. The third row is optimizing docking score against the M$^{\text{pro}}$ of SARS-Cov-2.

# I  HYPERPARAMETER TUNING

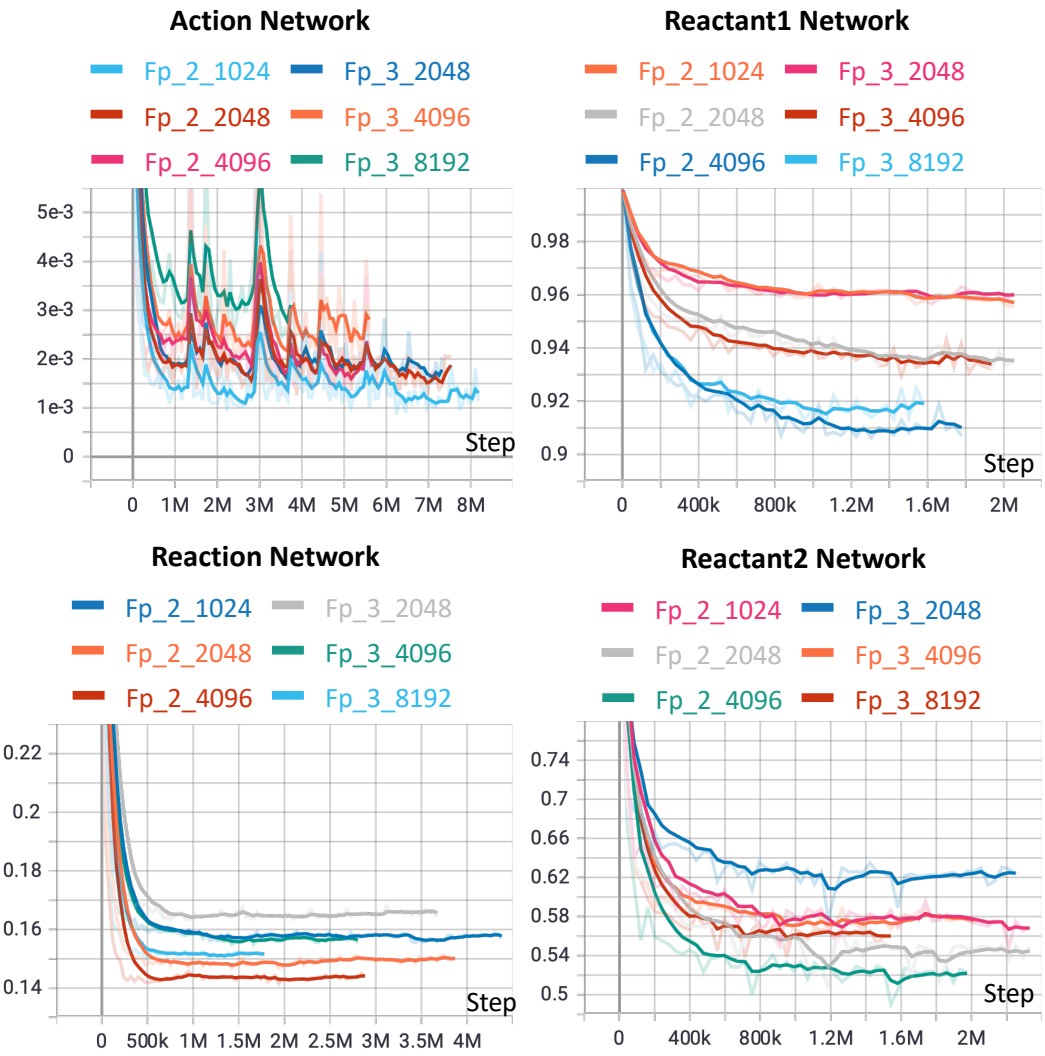

Figure 13: The validation loss during training with different radius and number of bits as network input. The validation loss is 1−accuracy, where the accuracies for the reactant networks are the accuracies of the k-NN searches ($k = 1$).

Besides Morgan fingerprints, other molecular representations explored during hyperparameter tuning included Graph Isomorphism Network (GIN) (Xu et al., 2018) embeddings and RDKit 2D descriptors (Figure 14). Morgan fingerprints of length 256 and radius 2 were found to work best.

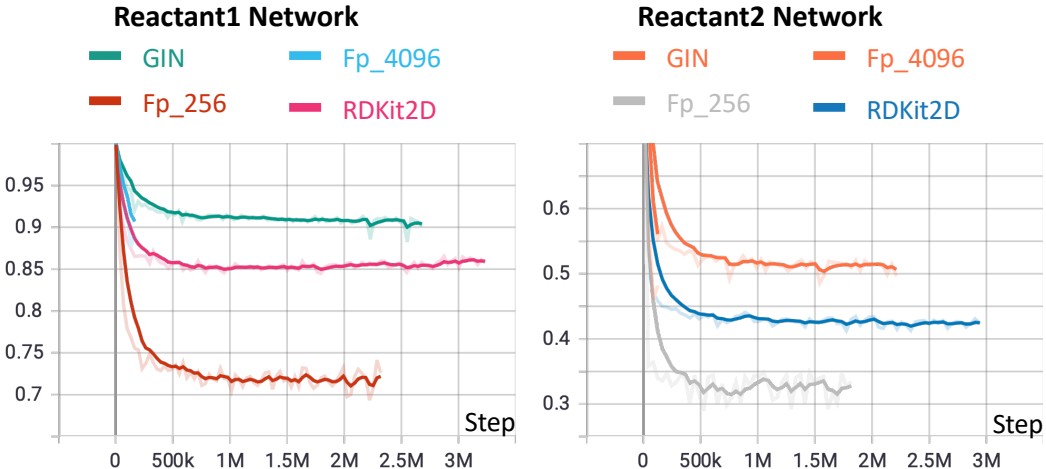

Figure 14: The validation loss during training using different action embeddings to conduct the k-NN search. The validation loss is $1-$accuracy, where the accuracies for the reactant networks are the accuracies of the k-NN searches ($k = 1$).

