# OpenReview forum: "Amortized Tree Generation for Bottom-up Synthesis Planning and Synthesizable Molecular Design"
_ICLR.cc/2022/Conference — ICLR 2022 Spotlight_

### Official Review · Reviewer_JJJC · 2021-10-27

**Correctness:** 2
**Technical Novelty And Significance:** 3
**Empirical Novelty And Significance:** 2
**Recommendation:** 3
**Confidence:** 4

**Main Review:**

## Strengths
### + Reasonable model of synthetic pathway
The proposed generation model enables us to use not only the purchasable molecules but also intermediate molecules to chemical synthesis whilst the model being simple enough.

## Weaknesses
### - The relationship between the proposed method and existing forward-synthesis-based methods is not clear
I am not very satisfied with the discussion on the relationship to the existing methods. The introduction section does not mention the existing work using the same forward-synthesis idea to construct molecules, and the relationship is discussed only in Section 2.1. In addition, the existing synthesizable molecular design methods are not compared in the experiments. These are just examples of a lack of connection to the existing literature. I would suggest the authors to strongly relate the proposed method to the existing work and discuss what are the contributions of the present work.

## Other suggestions
- In Section 3.3, the agent uses $k$-NN to choose reactants, but it is not clear how to use $k$ neighbors to select one reactant. It is in fact discussed in Section 4.2, but I suggest to explain it in the method section.
- Is it possible to select a reaction template first, instead of a reactant? The number of reaction templates is generally smaller than that of reactants, and it would be helpful to alleviate the initial reactant selection problem discussed in Section 5.

**Summary Of The Paper:**

The present paper is concerned about a forward-synthesis approach to molecular optimization. The proposed method defines one action to halt the procedure and three actions to develop a synthetic pathway, each of which corresponds to using zero/one/two of the intermediate molecules for chemical reaction to yield the next intermediate molecule. This generation model is controlled by an agent that chooses a series of actions to transit to the next state, given an embedding vector. If applied to synthesis planning, the embedding of the target molecule is used as the embedding vector, while if applied to molecular optimization, the embedding vector is optimized by a genetic algorithm.

The effectiveness of the proposed method is demonstrated by two experiments. The first one examines the ability of synthesis planning using reachable and unreachable data sets. The second one examines the ability of molecular optimization. For both of the experiments, both quantitative and qualitative analyses are provided.

**Summary Of The Review:**

I suggest to reject this paper and encourage the authors to revise it for next opportunities.

While I consider the method proposed in this paper is reasonable, the relationship to the existing work is not discussed enough, and therefore, I consider the paper is not very mature to be published as an academic paper. I would suggest the authors to revise the paper so as to clarify the difference from the existing work and how much the difference contributes to improve the performance. To do so, additional experimental results would be necessary.

---

> ### Author Response · Authors · 2021-11-22
> **We have added a comparison between previous methods and additional explanation about method.**
>
> Thank you for your comments. Please find our response below. We have also updated the draft and highlighted the modified part.
>
> ---
> REVIEWER COMMENT
> ---
>
> Relation to prior work
>
>
> AUTHOR RESPONSE
> ---
>
> Thank you for the comment. We have updated the introduction and related work sections to detail the difference between our work and previous work. Please see the summary comment above. To your comment about discussing related work in the Related Work section v. the introduction, we have also added a sentence in the introduction to be explicit that this task existed as a task prior to this manuscript.
>
> ---
> REVIEWER COMMENT
> ---
> How to use k-NN is not clear.
>
>
> AUTHOR RESPONSE
> ---
>
> Thanks for the comment. We use the predicted embedding as a query to pick the nearest neighbor among the building blocks. We have added a more detailed explanation in section 3.3 together with a reference to related method [1].
>
> ---
> REVIEWER COMMENT
> ---
> Is it possible to select a reaction template first?
>
>
> AUTHOR RESPONSE
> ---
>
> Thanks for this constructive comment. It is possible to select a reaction template first in the “Add” action, but not in others. We agree that may help alleviate the first reactant selection problem. We have added a note about this to this section, which was moved to the Appendix due to length constraints.
>
> ---
>
> Reference:
>
> [1] T. Cover and P. Hart, "Nearest neighbor pattern classification," in IEEE Transactions on Information Theory, vol. 13, no. 1, pp. 21-27, January 1967, doi: 10.1109/TIT.1967.1053964.

---

> > ### Comment · Reviewer_JJJC · 2021-11-24
> > **Re: Author response**
> >
> > Thanks for the response.
> >
> > ## Relation to prior work
> > As the authors mentioned in the introduction as "synthesizability-constrained generation has emerged as a promising alternative to this two-step pipeline", molecular generation using synthetic pathways is valuable because it allows us to avoid from synthesis planning. Then, I wonder why the authors still care about synthesis planning and why both synthesizable molecular design and synthesis planning have to be solved by the same method.
> >
> > In addition, if the present work is positioned as an improvement over some existing studies, then the authors have to demonstrate at least empirically that the proposed method is better than the existing ones quantitatively.
> >
> > In summary, I wish the authors clarify:
> > 1. Why do we have to care about synthesis planning?
> > 2. Why does one method have to solve two separate problems, synthesizable molecular design and synthesis planning?
> > 3. Empirical comparison with retrosynthetic tools in a synthesis planning scenario.
> > 4. Empirical comparison with synthesizable analog recommendation tools in a recommendation scenario.
> > 5. Empirical comparison with synthesizable molecular design tools in a molecular optimization scenario.
> >
> > ## $k$-NN
> > Thanks for the update. What I suggest is to clarify how to choose __one__ reactant $a_{\mathrm{rt1}}^{(t)}$ from $k$ nearest neighbors. In other words, $k$-NN in the right-hand side of Eq. 1 is expected to return $k$ objects, but we have only one output $a_{\mathrm{rt1}}^{(t)}$ the left-hand side, which is confusing to me.

---

> > > ### Author Response · Authors · 2021-11-28
> > > **Additional clarifications 1**
> > >
> > > We appreciate the opportunity for this additional discussion and address each item in turn. Please note, however, that making quantitative comparisons between different methods is deceptively simple given that the overall performance of a synthesizability-constrained approach cannot always be condensed to a single scalar value.
> > >
> > > **Why do we have to care about synthesis planning?**
> > >
> > > >Synthesis planning is required to physically produce molecules for the purposes of either (a) validating computational designs in a discovery setting or (b) manufacturing the molecule in a production setting. We would refer the reviewer to an article by industrial practitioners for a commentary on its importance [1]. In the manuscript, we have summarized the importance of synthesis planning in the first sentence of the third paragraph: “Planning and executing a practical synthetic route for a hypothetical molecular structure is a bottleneck that hinders the experimental validation of molecular design algorithms”
> > >
> > > **Why does one method have to solve two separate problems, synthesizable molecular design and synthesis planning?**
> > >
> > > >Treating the problems separately is the post hoc filtering strategy to which we refer in the third paragraph of the introduction. It is impractical to use this option, as quantitatively shown by [2], because for certain combinations of de novo methods and optimization objectives, it may be the case that none of the top-100 recommendations are synthesizable. Combining the two tasks could avoid additional computational cost (requirement of tens of seconds or minutes to plan a route to one target) and increase the success rate of molecular design. Besides, our method presents the first successful example of a bottom-up approach that has the potential to be more computationally efficient by mitigating the need for a tree search. This approach opens a new venue for synthesis planning as conditional generations, which potentially leads to a novel class of rapid synthesis planning methods and largely expands the utility of synthesis planning. We will add a sentence to this discussion in the introduction to more fully explain this.
> > >
> > >
> > > **Empirical comparison with retrosynthetic tools in a synthesis planning scenario**
> > >
> > > >This point is where comparisons resulting in a single quantitative metric become impossible. There are multiple ways to examine synthesis planning models, but no fair quantitative metrics exist for head-to-head comparisons of multi-step planning, even between retrosynthetic tools. Each “tool” will have a different notion of what chemical reactions are feasible, typically based either on a set of reaction templates or a learned template-free model. None will perfectly describe physical reality. Additionally, they will have different notions of what starting materials are commercially available.
> > >
> > > >The most sensible comparison between our approach and a retrosynthetic tool is implied by the “Reachable” row in Table 1. The test set for our model comprises target molecules for which we know synthetic pathways exist according to the set of building blocks and set of templates used in this study. Therefore, a retrosynthetic program with an infinite computational budget would be able to identify routes for all 69,548 reachable molecules (a recovery rate of 100%, average similarity of 1.0). In practice, with finite computational resources, retrosynthetic programs explore only a tiny fraction of the combinatorial space of possible retrosynthetic disconnections, but the extent to which performance would degrade from 100% recovery depends on how the tree search is implemented. As mentioned in other responses to reviewer 1rvZ, the purpose of our study is not to evaluate tree search algorithms.
> > >
> > > **Empirical comparison with synthesizable analog recommendation tools in a recommendation scenario**
> > >
> > > >This is a relatively new task/application. The definition of “analog” is imperfectly captured by the average similarity metric used in Table 1 for “unreachable” molecules. RetroDoG and MoleculeChef do not report their average similarity quantitatively, except  a few examples in RetroGoG’s Appendix (Figures 16, 17, 18). A systematic evaluation was not performed beyond these three examples.  The considerations of how allowed (“chemically reasonable”) disconnections are defined and how the library of starting materials is defined are also relevant to this situation.
> > >
> > >
> > >
> > > [1] Cornwall, Philip, Louis J. Diorazio, and Natalie Monks. "Route design, the foundation of successful chemical development." Bioorganic & medicinal chemistry 26.14 (2018): 4336-4347.
> > >
> > > [2] Gao, Wenhao, and Connor W. Coley. "The synthesizability of molecules proposed by generative models." Journal of chemical information and modeling 60.12 (2020): 5714-5723.

---

> > > > ### Comment · Reviewer_vmGs · 2021-11-28
> > > > **i don't think further empirical studies really add much to this paper, it is already good enough**
> > > >
> > > > I believe an empirical comparison to retrosynthetic tools is out of scope for this paper and can be addressed in future work, in particular given the short reviewing cycle for ML conferences. It would also be a not very meaningful comparison, as the authors explained, and also because most of these tools perform tree search with the aim to find multiple routes. It would be clear a priori that the synthesis planning algorithm would be slower than the model presented in this paper.
> > > >
> > > > *Empirical comparison with synthesizable analog recommendation tools in a recommendation scenario.*
> > > > Since this point would need a lot of domain expertise to evaluate, I also believe this is point is out of scope of this paper at an ML conference, and can be addressed in future work in a chemistry focussed venue.
> > > >
> > > > *Empirical comparison with synthesizable molecular design tools in a molecular optimization scenario.*
> > > > - this has now been provided via the guacamol results
> > > >
> > > >
> > > > I believe the paper does not need further experiments, it is significant enough and can be accepted now.

---

> > > > > ### Comment · Reviewer_JJJC · 2021-11-29
> > > > > **Re: Reviewer vmGs**
> > > > >
> > > > > Thank you very much for the comment. I am still not for accepting this paper.
> > > > >
> > > > > Since one of the contributions of this paper is the rapid bottom-up synthesis planning that has relatively high reconstruction accuracy, the authors have to demonstrate how rapid it is and how accuracy it is. These performance measures are relative; in other words, without any reference value, one cannot tell whether the proposed method is rapid or accurate. Therefore, I would expect the authors to provide such reference values. In the synthesis planning scenario, such a reference value could be the performance of retrosynthetic tools, and thus, I asked the authors to compare the performance with them.

---

> > > > ### Comment · Reviewer_JJJC · 2021-11-29
> > > > **Re: Additional clarifications 1**
> > > >
> > > > Thank you very much for the response.
> > > >
> > > > Regarding the first two questions, we may argue on different planes. Let me state my point in another way. In order to obtain an optimized molecule with its synthetic pathway, there are two approaches.
> > > >
> > > > 1. Run any molecular optimization method to obtain an optimized molecule, and then, run a synthesis planning tool to obtain a synthetic pathway to the molecule.
> > > > 2. Run a molecular optimization method that can output not only an optimized molecule but also its synthetic pathway.,
> > > >
> > > > This paper proposes a method based on the second approach, but also argues that their method can be used as a synthesis planning tool, implying that it can be used also for the first approach. Then, which approach is better? If the second approach is better than the first one, it is not important to develop a synthesis planning tool (at least for the purpose of molecular design); if the first approach is better than the second one, it does not make sense to take the second approach. Of course I understand it is difficult to give a conclusion to this question, but it looks weird to me to parallelize these two approaches in one paper. I would suggest to restructure the paper to focus on either "molecular optimization with synthetic pathways" or "synthesis planning".
> > > >
> > > > > Empirical comparison with retrosynthetic tools in a synthesis planning scenario
> > > > I understand a potential benefit of the proposed method over the conventional retrosynthetic tools is the computation time. Then, why not directly compare the computation time? A plot with computation time on the x-axis and recovery rate on the y-axis will demonstrate the benefit, and given a typical time constraint in a molecular design task, one can tell which method achieves better recovery rate; in contrast, without such an empirical comparison, the benefit is just a conjecture, not a fact.

---

> > > ### Author Response · Authors · 2021-11-28
> > > **Additional clarifications 2**
> > >
> > > **Empirical comparison with synthesizable molecular design tools in a molecular optimization scenario**
> > >
> > > >Again, the considerations of how allowed (“chemically reasonable”) disconnections are defined and how the library of starting materials is defined are also relevant to this situation. Here, we briefly resummarize the methods used by each and the prospect of a direct comparison:
> > > - MoleculeChef : one-step only, template-free : Quantitative results are only for distribution learning, not goal-directed optimization. The only optimization experiment is displayed in Figure 4, which shows a shift toward higher QED values, but other oracle functions are not evaluated.
> > > - ChemBO : multi-step, template-free : In their Table 3, they show that the maximum QED score achieved (though only after 100 evaluations) is 0.941, worse than the 0.948 achieved by most generative models including our own. Running ChemBO’s open source code on other objectives yields JNK (0.648), and GSK (0.492), which are substantially worse than ours: JNK: 0.719, GSK: 0.815. All these numbers are top-100 averages.
> > > - PGFS : multi-step, template-based, linear only : PGFS is closed source and only reports quantitative optimization results in their Table 2. QED optimization achieves the expected maximum score of 0.948. Their other tasks “RT”, “INT”, and “CCR5” are oracle functions taken from surrogate QSPR models which are not made available.
> > > - REACTOR : multi-step, template-based, linear only : The optimization evaluation in REACTOR (Table 1) focuses on the DRD2 oracle function; a mean activity of only 0.77 is achieved. Our model, as shown in our Table 7, identifies molecules with a score of 1.0 and 100 molecules with an average score of 0.998, which is far superior.
> > > - DoG-Gen : multi-step, template-free : The Appendix presents the results of optimization for 10 GuacaMol tasks in terms of top-1 score. We have run additional experiments with our model on the GuacaMol tasks to enable comparison with Table 7, with the significant caveats that the definitions of building blocks and reactions (template-based v. template-free) are not the same. In general, our model’s performance is comparable to DoG-Gen’s: ours perform at least as good as theirs in four among nine tasks. However, as template-free reaction models are trained only on successful reactions, they consistently predict a product even if the reactants cannot react, which leads to unreliable and overoptimistic reactions that would fail experimentally. Adopting such a model enlarges the chemical space it explores at risk of including unsynthesizable molecules, which leads to higher objective scores but physically unfeasible design. Besides, please note this is DoG-Gen, but not RetroGen. DoG-Gen is a specialized molecular optimization model and cannot perform synthesis planning and synthesizable analog recommendation tasks.
> > >
> > >
> > > | Methods | DoG-Gen [3] |	SYNOPSIS [3]	| Ours (top-1)	| Ours (top-100) |
> > > | --- | --- | --- | --- | --- |
> > > | Amlodipine |	0.80 |		0.63 | 		0.81 |	0.79 |
> > > | Aripiprazole | 	1.00 |		0.87 | 		0.63	|	0.59 |
> > > | Deco Hop |	1.00 |		0.88 | 		0.61	|	0.58 |
> > > | Osimertinib | 	0.89 |		0.84 | 		0.65	|	0.62 |
> > > | Perindopril | 	0.70 |		0.55 | 		0.68	|	0.61 |
> > > | Scaffold Hop | 	0.67 | 	0.54 | 		0.81	|	0.80 |
> > > | Sitagliptin | 	0.55 	|	0.43 | 		0.71	|	0.61 |
> > > | Valsartan | 	1.00 	|	0.00 | 		0.72	|	0.69 |
> > > | Zaleplon | 	0.62 	|	0.52 |		0.62	|	0.61 |
> > >
> > >
> > > [3] Bradshaw, John, et al. "Barking up the right tree: an approach to search over molecule synthesis dags." NeurIPS (2020).
> > >
> > >
> > > **k-NN clarification**
> > >
> > > >To select only one reactant, we set $k=1$, i.e., only the immediately nearest neighbor is used. For $k=3$ as used for the first reactant search in 4.2, the selection of three possible “first reactants” is done analogously to a beam search, where three full synthetic trees are generated thereafter, each using one of the choices. From those three trees, which generate up to three unique product molecules, we choose the one that is the most similar to the target molecule as reflected by the Tanimoto similarity using Morgan fingerprints. We will clarify this in the camera ready version but can no longer post revisions during the review period.

---

> > > > ### Comment · Reviewer_vmGs · 2021-11-28
> > > > **great results!**
> > > >
> > > > Dear Authors,
> > > >
> > > > thank you these additional results on Guacamol, they are very impressive for a reaction driven de novo design algorithm.
> > > >
> > > > Dear fellow Reviewer JJJC,
> > > >
> > > > These results now quantitatively demonstrate:
> > > > a) similar performance on optimisation tasks to state of the art reaction-driven de novo design algorithms
> > > > b) the first thorough investigation of conditional synthesis tree generation which quantitatively good reconstruction performance, which on is own is a major milestone
> > > >
> > > > I would suggest to reconsider your rejection of the paper now, and to accept the paper.

---

> > > > ### Comment · Reviewer_JJJC · 2021-11-29
> > > > **Re: Additional clarifications 2**
> > > >
> > > > I would appreciate the authors to provide additional comparisons. These are (though not complete) very helpful to understand the capability of the proposed method. If the authors could show real examples of the failure of the template-free methods, the comparison with DoG-Gen would become very convincing, even if the performance of the proposed method is comparable or sometimes worse than DoG-Gen.

---

> > > > > ### Comment · Reviewer_vmGs · 2021-11-29
> > > > > **template based vs free will not tell you much in this case**
> > > > >
> > > > > I don't think the comparison between template-based and free methods will not add much, for two reasons:
> > > > >
> > > > > 1) The comparison between template-based and template-free methods is an active research topic on its own (and out of scope for this paper), which I think cannot be asked from the authors in the short discussion period. There are also no clear quantitative benchmarks to measure the quality of the proposed reactions.  This should be conducted at a chemistry journal, with reviewers with an organic chemistry background (which not so many reviewers at ICLR have).
> > > > >
> > > > >
> > > > > 2) Dog-Gen explicitly can also be used with template-based models, or more sophisticated reaction prediction pipelines as described in Segler el al Nature 2018 or Coley et al Science 2019. Similarly, such pipelines could be added to the model presented here. A template-free vs template-based analysis would thus not tell much about the capabilities of Dog-Gen. That the optimisation performance is similar to Dog-Gen has already been demonstrated on the Guacamol Benchmarks. The authors followed established practices in the validation of optimisation.
> > > > >
> > > > > Also, the focus of the current paper here is getting conditional tree generation to work properly. This has been achieved. This has also not been demonstrated for dog-gen/retrodog. That the model also performs decently on optimisation is a plus, but not the main point of this paper.
> > > > >
> > > > > I would thus encourage you to now accept the paper, and leave the other experiments for future work. Papers don't have to be perfect, just good enough :)

---

> > > > > > ### Comment · Reviewer_JJJC · 2021-11-30
> > > > > > **Re: Reviewer vmGs**
> > > > > >
> > > > > > While I agree that a paper does not have to be complete, I consider a paper has to describe its contributions consistently and correctly. My concerns are twofold. First, I don't still understand why the proposed method, "eliminating the need for two-stage pipelines of generation and filtering" (cited from the conclusion section), should be capable of rapid bottom-up synthesis planning, which is mainly used for filtering; thus I feel the paper is not very consistent. Second, the contributions listed in the introduction were not well-supported by empirical results (which was partially improved by the recent GuacaMol benchmark).
> > > > > >
> > > > > > If these two of my concerns are not significant or wrong, I don't oppose to accept this paper. I would like to ask the ACs to judge it.

---

### Official Review · Reviewer_1rvZ · 2021-10-31

**Correctness:** 3
**Technical Novelty And Significance:** 3
**Empirical Novelty And Significance:** 3
**Recommendation:** 8
**Confidence:** 3

**Main Review:**

The advantages of this paper are as followings:
1. The idea of constructing a synthetic tree in a bottom-up manner to solve both problems of molecular design and synthesis planning simultaneously is interesting and eye-catching. The tree generation algorithm is succinct and efficient.
2. Case studies(figure 5, figure 11) demonstrate the results in details.
3. It’s nice to see hyperparameter tuning results.
4. Codes are available.

Concerns are as followings:
1. It seems not very clear to me that how is genetic algorithm integrated into the process of molecular optimization and synthesis planning.
2. It might be better to compare the recovery rate(shown in table 1) of the proposed method with other synthesis planning methods(i.e., retrosynthe- sis).
3. It might be valuable to analyse more about why the proposed method is more computationally efficient.
4. In table 2, it would be better to demonstrate quantitatively the synthetic accessibility of molecules generated by different methods.

Some typos:
1. The font size of table 6 is too large.


**Summary Of The Paper:**

This paper aims to construct a synthetic tree in a bottom-up manner to solve both problems of molecular design and synthesis planning simultaneously. The root node represents the final generated molecule, and the leaf nodes represent the reactants that can be purchased. Starting from the bottom leaf nodes, the authors utilize reaction templates to generate the parent node until the root node is reached. To design molecules with desired properties, the authors utilize the genetic algorithm to guide the tree generation.

**Summary Of The Review:**

Overall, the paper is well-written, and the method appears to be promising.

---

> ### Author Response · Authors · 2021-11-22
> **We have added a comparison between previous methods and added more explanation about the genetic algorithm.**
>
> Thank you for your comments. Please find our response below. We also update the draft and highlight the modified part.
>
> ---
> REVIEWER COMMENT
> ---
>
> It is not clear how genetic algorithm is integrated
>
>
> AUTHOR RESPONSE
> ---
>
> Thanks for the comment. We have added an illustration of the genetic algorithm and additional explanation in Appendix B of the updated draft. We use the conditional synthetic tree generator as a decoder to obtain molecules corresponding to input fingerprints. We crossover and mutate on the pool of fingerprints to optimize the molecule implicitly. For synthesis planning the conditional code corresponds to a proper molecular fingerprint generated for the structure, but during optimization, the conditional code is simply treated as a boolean vector.
>
> ---
> REVIEWER COMMENT
> ---
> Compare recovery rate with other retrosynthesis models
>
>
> AUTHOR RESPONSE
> ---
>
> This is a very interesting point that requires a more nuanced discussion than it may seem. We agree a comparison with other synthesis planning tools would be valuable. However, given an unlimited amount of time, all synthesis planning algorithms, including ours if we allow infinite k, could find synthetic paths for the target molecules as long as they exist within the combinatorial chemical space defined by the set of reaction templates and set of building blocks, which leads to 100% recovery rates. Therefore, comparing to retrosynthetic models would be more about the comparison of starting materials and coverage of chemical reactions, which is not the goal of this paper, so we have chosen not to address this here. Note that our templates are in the forward synthetic direction, not the retrosynthetic direction, and are not trivial to invert without changing their implicit specificity.
>
> ---
> REVIEWER COMMENT
> ---
> Analysis of computational efficiency
>
>
> AUTHOR RESPONSE
> ---
>
> We conclude the computational efficiency is due to the amortized approach that doesn’t rely on explicit tree search processes. Even with k=1, we can recover one third of the test molecules. To our knowledge,  no published retrosynthesis program has ever reported results  a branching ratio of 1.
>
> ---
> REVIEWER COMMENT
> ---
> Demonstrate quantitatively the synthetic accessibility is better
>
>
> AUTHOR RESPONSE
> ---
>
> Thanks for the constructive comments. As far as we know, there is currently no better computational evaluation of synthetic accessibility than explicitly finding synthetic routes for the targets [1-3]. So given that all of our designs have synthetic routes to them, they are synthesizable under our definition (i.e., according to the set of templates, which we acknowledge are imperfect). Any quality filters or heuristic scores that agree with these results only show the metric is correlated with our definition of synthetic accessibility, but not the quality of our model. Nevertheless, we have included the SA_Score [4] and the percentage of molecules passing a heuristic filter of pharmaceutical quality [5] of the generated molecules to show the qualitative improvement in synthesizability as perceived by these commonly-used but flawed heuristics (Appendix H, Table 8).
>
> Reference:
>
> [1] Gao, Wenhao, and Connor W. Coley. "The synthesizability of molecules proposed by generative models." Journal of chemical information and modeling 60.12 (2020): 5714-5723.
>
> [2] Liu, Cheng-Hao, et al. "Retrognn: Approximating retrosynthesis by graph neural networks for de novo drug design." arXiv preprint arXiv:2011.13042 (2020).
>
> [3] Thakkar, Amol, et al. "Retrosynthetic accessibility score (RAscore)–rapid machine learned synthesizability classification from AI driven retrosynthetic planning." Chemical Science 12.9 (2021): 3339-3349.
>
> [4] Ertl, Peter, and Ansgar Schuffenhauer. "Estimation of synthetic accessibility score of drug-like molecules based on molecular complexity and fragment contributions." Journal of cheminformatics 1.1 (2009): 1-11.
>
> [5] Brown, Nathan, et al. "GuacaMol: benchmarking models for de novo molecular design." Journal of chemical information and modeling 59.3 (2019): 1096-1108.

---

### Official Review · Reviewer_GkiB · 2021-11-02

**Correctness:** 4
**Technical Novelty And Significance:** 3
**Empirical Novelty And Significance:** 3
**Recommendation:** 8
**Confidence:** 4

**Main Review:**


This is a well written paper that presents a sound, original idea.
I'm amazed that nobody had tested this idea before and I'm a little
surprised that it works as well as it did on a first try.
[EDIT: as reviewer "vmGs" pointed out, this idea is very closely
related to the NeurIPS2020 paper by Bradshaw et al, which I didn't
realize at the time of writing the rest of this review.  I will reassess
this work after any revisions, and have temporarily reduced
my previously enthusiastic score. EDIT2: the revisions addressed my concerns.]

The methodology exposition is clear. Perhaps an appendix with a
summary of the input/output dimensions of each trainable module would
help organize the scattered information about the sizes and types of
the various inputs and outputs and give a quick overview of the full
method (this could be as simple as an annotated version of Figure 2
with example dimensions from the best model.) Did the authors try to
input to the k-NN modules a (possibly pretrained) embedding of the
reagent graphs instead of the 256 Morgan Fingerprints of radius 2? And
did those Morgan fingerprints they used include the reagent chirality
information or not?

The results of the baseline models in Figure 5 and Figure 11 are at
best embarrassing for those other codes, and at worst some
misconfiguration of those codes. I imagine that the codes try hard to
design against a target fingerprint-based input function and just
throw the kitchen sink until most of the bits of the produced
molecules match something that improves the score in the target
function. Out of curiosity, did the authors of this paper try to see
if there were any "reasonable" molecules produced by the other codes
with score better or equal to that of their own model. In any case,
the message that the authors want to convey with the current examples
is clear, but it may be helpful to contact the authors of those other
papers as a fair warning.

The paper demonstrates the basic concept behind this work. I was left
wondering why the authors limited the selection of possible reaction
templates to such a small set---is there a practical reason related to
the performance of the model in terms of training or inference time?
Furthermore, why did the authors only limit the present work to k=1?
I appreciate the simplicity of documenting the greedy (k=1) selections
for the results in table 1, but I'm left wondering how much better
could these results become if one included a larger number of
candidates at each step and searched all relevant trees.

Although I commend the authors on including source code with the
submission, the reproducibility of this work is currently
unclear. Unfortunately, the supplemental code was not usable, so I
couldn't check any of the claims of performance. The README file
appears to be incomplete and has a couple of TODO notes in it; the
test subdirectory seems to be missing and thus the instruction to run
the unittests doesn't run any unittest at all; the code has references
to undeclared paths (DATAPATH, HOMEPATH, st_pis); references to data
files that don't exist (enamine_us.csv.gz).  I imagine that some of
these problems are easy to fix with a couple additional instructions
and it is possible that the rest could be resolved with a couple extra
scripts and some minor modifications.  Nevertheless, I felt a bit
disappointed that the authors missed an opportunity to submit a
remarkable work with a reproducible demonstration. I'd suggest that
the authors contact Enamine before the conference and request
permission to publish a snapshot of the current set of building blocks
or any other data that the authors used from Enamine, so that future
papers can compare against a benchmark using the same training and
testing data to document any algorithmic improvements in a bottom-up
approach or compare such approaches to retrosynthetic pathway
generation that target the same building blocks.

Despite these concerns on reproducibility, I think that this is a
refreshing paper of the type that I expect to see highlighted in the
ICLR conferences and I think that it opens up new directions for
methodological improvements to molecule optimization.


**Summary Of The Paper:**


This paper proposes a bottom-up approach to molecular synthesis
planning and synthesizable molecular design. The authors cast the
synthesis planning problem as a Markov Decision Process with trainable
networks for selecting next actions.  The output of the approach is a
binary tree that documents reaction and the combination of building
blocks or previously synthesized reagents that would mix to create a
new intermediate or a final molecule.  When the approach fails to
produce the correct target molecules it typically generates a similar
molecule---the authors take advantage of this feature of their
approach and optimize their algorithm to design synthesizable
molecules with desirable properties.


**Summary Of The Review:**


~This is a refreshingly original paper on molecular synthesis planning
and synthesizable molecular design.  It is a strong, foundational
paper that is well written. The authors could do some a little
work to improve the reproducibility and enable future benchmarks
against the same data.~

[Edit: this work closely relates to Bradshaw et al, but the relationship
was not made clear in the exposition.  Until the authors
address this concern and clarify their additional important contributions,
I do not recommend publication of this paper.]

Edit2: This is a strong paper that is well written and deserves publication at ICLR22.

---

> ### Author Response · Authors · 2021-11-22
> **We have added a comparison between previous methods and all code to reproduce the results are included.**
>
> Thank you for your comments. Please find our response below. We also update the draft and highlight the modified part.
>
> ---
> REVIEWER COMMENT
> ---
>
> Relation to prior work
>
>
> AUTHOR RESPONSE
> ---
>
> Thanks for the comments. We have updated the introduction and related work sections to detail the difference between our work and previous ones. Please see the summary comment above. If you feel that this is still insufficient contextualization, please let us know. We had mentioned prior approaches to similar tasks in the Related Work section  2.1 in the original draft and certainly did not mean to claim novelty in the task itself.
>
> ---
> REVIEWER COMMENT
> ---
> Hyper-parameter setting: small reaction template set and k=1 in present work
>
>
> AUTHOR RESPONSE
> ---
>
> We agree the reactions covered by our 91 templates is not exhaustive compared to the full scope of known synthetic organic chemistry h, but this quantity is actually sufficient to explore a very large chemical space. The Enamine REAL Space [1] comprises 21 billion synthesizable molecules that were generated by applying 170 reactions to 112,000 building blocks; while not disclosed, it is likely that that no more than 3 reaction steps were applied. The SAVI project [2] generated over one billion molecules with just 53 chemical transformations and 150,000 available building blocks. Previous template-based methods, such as PGFS [3], also use a smaller number of templates (64 unique reaction templates in PGFS). The combinatorial nature of multi-step synthesis planning means that 91 templates and ca. 100,000 building blocks can access billions and billions of molecules even with ≤5 reaction steps.
>
> The setting of k to a small number is for evaluation purposes. We want to evaluate how well the neural networks are trained, but not how well a tree search algorithm works. If we adopt k > 1, we do see  better recovery results as a result of exploring more branches in the search tree, but at the cost of a longer search. The combinatorics of multi-step planning also means that using a large k becomes impractical for deep searches containing many reaction steps. This is a parameter that can be tuned by the users depending on their goals.
>
> ---
> REVIEWER COMMENT
> ---
>
> Reproductivity
>
>
> AUTHOR RESPONSE
> ---
>
> Thanks for the comment. We have included all of the source code to reproduce the results in our initial submission. The path names were changed for this anonymous submission and one just needs to replace the paths into their local path to the code and data. The Enamine building blocks data can be downloaded from https://enamine.net/building-blocks/building-blocks-catalog freely for academic purposes. We have included all instructions in the README, and have also made minor updates to the code since the original submission to address incomplete TODOs and updated instructions to make it clearer how to run the code and where to get the data. The repository associated with this project will be linked to in the manuscript upon publication and contains significantly more detail about reproducibility and use than the anonymized supplemental submission here.
>
>
> Reference:
>
> [1] https://enamine.net/compound-collections/real-compounds/real-space-navigator
>
> [2] Patel, Hitesh, et al. "SAVI, in silico generation of billions of easily synthesizable compounds through expert-system type rules." Scientific data 7.1 (2020): 1-14.
>
> [3] Gottipati, Sai Krishna, et al. "Learning to navigate the synthetically accessible chemical space using reinforcement learning." International Conference on Machine Learning. PMLR, 2020.

---

> > ### Comment · Reviewer_GkiB · 2021-11-23
> > **Thank you for the updated manuscript.**
> >
> > The authors have addressed my sole important concern about this work in their revision and I have thus reverted to my original score of 8. I strongly believe that this work deserves publication at ICLR22.  I hope that the addition of new templates and improved ways to explore the possible reaction trees will further improve this direction of research in the near future.

---

### Official Review · Reviewer_vmGs · 2021-11-03

**Correctness:** 4
**Technical Novelty And Significance:** 3
**Empirical Novelty And Significance:** 4
**Recommendation:** 8
**Confidence:** 5

**Main Review:**

## Strengths:
- interesting extension of the work by Bradshaw, Gottipatti and Horwood, which addresses some of the limitations of these models.
- first in-depth investigation of conditional synthesis dag generation with compelling results (this was studied on a very shallow level by Bradshaw et al)
- interesting observations which contrast prior work: Using fingerprints instead of GNNs seems to be beneficial, and the use of a genetic algorithm on binary fingerprints seems to be at least competitive with optimisation in a continuous latent space.
- reasonable experiments in line with prior work
- good results in comparison with other, less restricted molecule search algorithms

## Weaknesses:
### Relation to prior work
Upon reading the related work in more detail, it is clear that this work is an extension of the work by Bradshaw et al. 2020 NeurIPS, who reported 1) generation of synthesis trees (synthesis trees are a special case of DAGs) using a generative model for DAGs, and 2) conditional tree generation $p(T,x)$. Conditioned on an embedding of the desired product structure $x$, the generative model generate a synthesis tree $T$. This model is called "RetroDOG". Furthermore, the MDP setting has been explored by Horwood and Gottipatti.

To be more concrete: In section D.3 of the appendix https://papers.nips.cc/paper/2020/file/4cc05b35c2f937c5bd9e7d41d3686fff-Supplemental.pdf  of their Neurips paper, Bradshaw et al. write:
`Our generative model of synthesis DAGs can be seen as a parametrizable mapping from a vector of real numbers to a synthesis DAG. As a module it can be mixed and matched in different ML frameworks as we have already seen in the main paper with DoG-AE and DoG-Gen. In this section we describe some preliminary results with a third model architecture, called RetroDoG. This model consists of the composition of a GNN followed by our generative synthesis DAG model to produce a learnable mapping from a molecular graph to a synthesis DAG. By training this model on pairs of product molecules and their associated synthesis DAGs we can use this model to perform retrosynthesis (i.e. predict how a particular product can be made). [...] the model described in this section would additionally allow one to feed in a potentially hard or impossible to synthesize molecule, and obtain a similar molecule which is easy to synthesize, which is impossible with current planners." [...] "RetroDoG in contrast tries to construct the DAG in a **bottom-up** manner [...].  we believe such an approach as RetroDoG may be worthy of future research interest, and may for instance be useful in combination with more complex tools to **amortize** and reduce the cost of **searching for synthetic routes**."


Building on other people's work is an essential part of science, but it needs to be acknowledged accordingly, and there is no harm in doing so. In fact, the bigger harm to the scientific community comes from not doing so.

Unfortunately, Bradshaw et al. 2020, Gottipatti et al and Horwood et al are only cited in passing in the related work section, and not in the introduction, and there is not even a mention of the RetroDog model by Bradshaw. In the way the contributions of this paper here are presented, it leads to the impression that they were actually invented by the authors, which is not the case.

`We formulate the tasks of multi-step synthesis planning and synthesizable molecular design as a single shared task of conditional synthetic tree generation. ` => this has already been done by Bradshaw et al as the RetroDog model

`We formulate a Markov decision process to model the generation of synthetic trees, allowing the generation of multi-step and convergent synthetic pathways.` => Gottipatti & Horwood used the MDP framing already

`We propose a model that is capable of (1) rapid bottom-up synthesis planning and (2) constrained molecular optimization that can explore a chemical space` => this is the RetroDog model by Bradshaw et al (who also explicitly did not restrict their model to the molecular transformer).


The paper here should thus be clearly presented as an investigation and extension of the work by Bradshaw, Gottipatti and Horwood, rather than novel modelling. This would not detract from its quality or value, which in my opinion is considerable.

This reviewer would also suggest to cite non-neural reaction-driven molecular design work, for example https://doi.org/10.1371/journal.pcbi.1002380 or https://doi.org/10.1021/jm030809x which predate generative models for molecules, and already point out the requirement of synthesizability. In fact, these algorithms have also been used in conjunction with fingerprint similarity to a target molecule as the objective, and would thus be a good baseline to explore in this work.


### Modelling
The model the authors describe should be more correctly referred to as a generative model for synthesis DAGs. There is no MDP here, since there is no notion of reward.

Also, it can be argued that a synthesis tree does not really satisfy the Markov assumption. For example, one would never introduce a protecting group A, and then immediately deprotect A.

### Experiments and Baselines
Most if not all molecules that come from the baselines do not appear to be very drug-like. This raises the question whether the used benchmark functions are really realistic enough for drug discovery, or whether other objectives should be considered.
Furthermore, are cLogP or QED really a quantity that is desirable as an objective for drug discovery on its own? Also, are docking scores really reliable enough to be used a quantitative oracle?




**Summary Of The Paper:**

The authors explore an interesting extension of Bradshaw's generative model for synthesis trees, which leads to results for molecule optimisation and generation, which are competitive to less restricted models.

**Summary Of The Review:**

Updated after Author response:

I think the paper should now be accepted at ICLR. It is a significant step forward for the field, with a modeling approach interesting from the ML side as well as non-trivial, domain-relevant empirical validation (which many other papers on the topic at ML conferences do not have).



nit: with regards to the baseline experiments side: I would suggest to move the QED task to the appendix.

---

> ### Author Response · Authors · 2021-11-22
> **We have added a comparison between previous methods. The oracle functions are adopted for a benchmarking purpose.**
>
> Thank you for your comments. Please find our response below. We also update the draft and highlight the modified part.
>
> ---
>
> REVIEWER COMMENT
> ---
>
> Relation to prior work
>
> AUTHOR RESPONSE
> ---
>
> Thank you for the comments. We have updated the introduction and related work sections to detail the differences between our work and previous publications. Please see the summary comment above.
>
> ---
> REVIEWER COMMENT
> ---
> Modeling: There is no MDP in our work because no notion of reward, and Markov properties are not satisfied.
>
>
> AUTHOR RESPONSE
> ---
>
> We feel our formulation is accurately described as an MDP, where the reward is inherently defined by the task purposes: for synthesis planning, the reward is the similarity of the product to the target molecule, with a similarity of 1.0 being the highest reward and indicating a perfect match; for molecular design, the reward is determined by how well the product properties match the desired criteria. For clarity, we have mentioned the reward definition explicitly in the updated draft. While this reward is not needed during model training, it can be used during optimization as the fitness for the genetic algorithm.
>
> The Markov property is naturally satisfied by the bottom-up formulation: upon obtaining a specific compound (an intermediate in a synthetic route), subsequent reaction steps can be inferred entirely from that intermediate compound, and do not depend on the pathway used to get to said compound. In your example, one just needs to know if the functional group is protected or not (e.g., has A or not) to proceed to further steps. The optimal strategy should never be to protect and immediately deprotect, so the Markov assumption is preserved when we consider the optimal policy conditioned on the target compound we are building the pathway towards.
>
> ---
>
> REVIEWER COMMENT
> ---
> Experiments and Baselines
>
> AUTHOR RESPONSE
> ---
>
> The oracle functions we adopted, such as cLogP, QED, and docking scores, are estimations of pharmaceutically-relevant properties that are widely used in industrial drug development processes such as virtual screening [1]. We absolutely recognize that in practical discovery workflows, they are not optimized in isolation but used in combination with other properties in a multi-objective setting. However, none of these computational oracle functions are perfect and may suffer from poor generalization to less drug-like molecules, which leads to the design of false positives. This is a common problem faced by all de novo design methods, and in fact is one of the motivations for this work:  we need to constrain the design space to a synthesizable chemical space rather than simply a (syntactically) valid chemical space.
>
> We agree that the benchmarking of methods for molecular optimization is not directly aligned with realistic experiments used in drug discovery. This is something that the field has struggled with for years. cLogP and QED have dominated performance comparisons until quite recently, when docking scores were introduced as slightly more challenging (but still idealized) objectives. Developing a robust set of benchmarking tasks is not an intended contribution of this work. The aim of this work is to present a novel approach to synthesizable molecular design and demonstrate its empirical success, not to investigate a specific application case of each oracle function. For lack of a better (computational) approach, we need to adopt these commonly used oracle functions as baselines to verify our model’s optimization ability and compare with other methods [2-5].
>
> Reference:
>
> [1] Bender, Brian J., et al. "A practical guide to large-scale docking." Nature Protocols (2021): 1-34.
>
> [2] Jin, Wengong, Regina Barzilay, and Tommi Jaakkola. "Multi-objective molecule generation using interpretable substructures." International Conference on Machine Learning. PMLR, 2020.
>
> [3] Jin, Wengong, Regina Barzilay, and Tommi Jaakkola. "Hierarchical generation of molecular graphs using structural motifs." International Conference on Machine Learning. PMLR, 2020.
>
> [4] Xie, Yutong, et al. "MARS: Markov Molecular Sampling for Multi-objective Drug Discovery." International Conference on Learning Representations. 2020.
>
> [5] Nigam, AkshatKumar, et al. "Augmenting Genetic Algorithms with Deep Neural Networks for Exploring the Chemical Space." International Conference on Learning Representations. 2019.

---

> > ### Comment · Reviewer_vmGs · 2021-11-22
> > **accept**
> >
> > Dear Authors,
> >
> > I want to thank you for the constructive discussion and have updated my review accordingly. The paper should be accepted now.
> >
> >
> > Best,
> > Reviewer vmGs

---

### Author Response · Authors · 2021-11-22
**Relation between previous methods**

Thank you for the constructive comments. We have updated the introduction and related work sections to detail the difference between our work and previous work. We have uploaded an updated version and highlighted the modified part.

We would like to first emphasize that there is tremendous scope for improving the practical trainability and utility of synthesizability-constrained models for molecular generation and exploring new implementations. We do not see this task as having in any way been fully explored by the few studies to date. This would be analogous to saying that after Gomez-Bombarelli’s CVAE paper in 2016, there was little novelty in any subsequent work using SMILES representations for de novo molecular generation, whereas there have been dozens or hundreds of contributions that have advanced the SOTA and have made these methods more useful in practice through a combination of machine learning and domain-specific contributions. We have clarified the contributions of our work relative to prior work, particularly Bradshaw et al. (2020), to not misrepresent the task novelty.

In short:
* We agree that the RetroDog model was the early attempt to this approach, **but it was unsuccessful and only applied for the synthesizable analog recommendation task (i.e., cases that fail to recover target molecules)**. We have mentioned it in the introduction and related work of the updated draft. We did not intend to underemphasize the foundation provided by Bradshaw et al. (2020), which was mentioned in the original draft but is now described in greater detail.
* Unlike ours, the MDPs formulated by Gottipatti & Horwood are limited to linear synthetic pathways, which means  **their models are unable to generate convergent pathways.** This formulation severely limits the chemical space the model can access, i.e., it cannot design many common drug molecules such as remdesivir, which we showed in Figure 6, Appendix A.
Bradshaw’s DoG-AE/GEN models are capable of synthesizable molecular design but the RetroDog mode is not capable of synthesis planning. Our model is the first successful attempt to amortized multi-step synthesis planning, meanwhile, it can be used for synthesizable molecular design.
* Additionally, Bradshaw’s DoG-AE/GEN/RetroDog requires a forward reaction predictor that operates on reactant molecules only, such as the Molecular Transformer (MT). Though it’s not restricted to the MT architecture, all of the machine learning forward predictors suffer from the bias of positive reaction data. As very few unsuccessful reactions were reported, all those models are trained only on successful reactions and will always predict a product even if the reactants cannot react, which leads to unreliable and overoptimistic reactions that would fail experimentally. The generalization of their model to adopting reaction templates is not at all trivial, as this introduces a discrete action space that significantly changes their problem formulation. Our framework bypasses the problem of positive bias by using a set of chemical transformations templates, which disallows reactions that do not match a known template. Reaction templates can also be refined to have arbitrary levels of specificity (e.g., as in the commercial SYNTHIA software program) which will increase the robustness of the overall pathway recommendations, whereas template-free models will always be susceptible to mispredictions and error propagation from the forward model.
* One of the distinctions that we feel allows our approach to be more successful is the use of intermediate molecule(s) returned by the environment to be included as input at each iteration, whereas Bradshaw et al. (2020) did not explicitly incorporate the structure of intermediate structures such that the model had to implicitly learn an approximation to the reaction product prediction model. We now mention this in the revised manuscript under “State Transition Dynamics”.

---

> ### Comment · Reviewer_vmGs · 2021-11-22
> **Thanks you for the clarification.**
>
> Dear Authors,
>
> I would to thank you for the clarifications.
>
> I believe the discussion of prior work is now appropriate.
>
> Also, to make it absolutely explicit, I consider this work as a major step forward for automated molecular design, with several important contributions on the modeling as well on the validation side.
>
> The paper should make its way into ICLR now.

---

### Decision · Program_Chairs · 2022-01-20

**Decision:**

Accept (Spotlight)

**Comment:**

After much back and forth about prior work, 3 reviewers score this paper as an 8 and one scores it as a 3.
Other reviewers have written to the 3 and told them they believe that their review is now too harsh, in light of clarifications w.r.t. related work. I tend to agree, though I must admit that I am not an expert on this topic.
Given that there is almost unanimous support for accepting and it's possible that the one hold-out has not seen some of the extra information, I recommend acceptance.
Given the praise from the other three reviewers, I moreover recommend a spotlight.